# Kolya: an open-source package for inclusive semileptonic $B$ decays

**Matteo Fael[1]⋆, Ilija S. Milutin[2]† and K. Keri Vos[3,4]‡**

**1** Theoretical Physics Department, CERN, 1211 Geneva, Switzerland
**2** Theoretische Physik 1, Center for Particle Physics Siegen,
Universität Siegen, D-57068 Siegen, Germany
**3** Gravitational Waves and Fundamental Physics (GWFP),
Maastricht University, Duboisdomein 30, NL-6229 GT Maastricht, the Netherlands
**4** Nikhef, Science Park 105, NL-1098 XG Amsterdam, the Netherlands

⋆ matteo.fael@cern.ch † Ilija.Milutin@uni-siegen.de ‡ k.vos@maastrichtuniversity.nl

## Abstract

We introduce the code `kolya`, an open-source tool for phenomenological analyses of inclusive semileptonic $B \to X_c l \bar{\nu}_l$ decays. It contains a library to compute predictions for the total rate and various kinematic moments within the framework of the heavy quark expansion, utilizing the so-called kinetic scheme. The library currently includes power corrections up to $1/m_b^5$. All available QCD perturbative corrections are implemented via interpolation grids for fast numerical evaluation. We also include effects from new physics parameterized as Wilson coefficients of dimension-six operators in the weak effective theory below the electroweak scale. The library is interfaced to CRunDec for easy evaluation of the quark masses and strong coupling constant at different renormalization scales. The library is developed in Python and does not require compilation. The code is flexible and can be extended to other kinds of inclusive decays such as $B$ meson lifetimes, $B \to X_u l \bar{\nu}_l$ and $D$ meson decays.

# 1 Introduction

Measurements of semileptonic $B$ decays lie at the core of the Belle II and LHCb physics programs in the upcoming years. Thanks to relatively large rates and clean experimental signatures, inclusive and exclusive semileptonic decays with a $b \to c\,l\,\bar{\nu}_l$ transition ($l = e, \mu$) offer a clean avenue for the determinations of $|V_{cb}|$, the element of the Cabibbo-Kobayashi-Maskawa matrix (CKM) which parameterizes the strength of the weak interaction among bottom and charm quarks in the Standard Model of particle physics.

    Inclusive determinations of $|V_{cb}|$ exploit that the semileptonic rate $\Gamma_{\text{sl}}$ and moments of kinematic spectra can be described with good precision using the heavy quark expansion (HQE) [1–3]. In the

HQE, these observables are expressed as a series of non-perturbative HQE elements proportional to increasing powers of the inverse bottom quark mass times the QCD scale parameter, $\Lambda_{\text{QCD}}/m_b$. In addition, each order in the HQE also receives corrections expressed as a series expansion in the strong coupling constant, $\alpha_s$, which can be systematically calculated in perturbative QCD.

This paper documents the first release of the open-source code `kolya` [4]. It consists of a Python library which computes the prediction for the total rate and lepton-energy, hadronic invariant mass and dilepton invariant mass kinematic moments within the framework of the HQE utilizing the so-called kinetic scheme [5–8]. The `kolya` code supersedes and extends the code developed for the fit of $q^2$ moments [9] measured by Belle [10] and Belle II [11]. We also include effects from new physics (NP) studied in Ref. [12]. These are parameterized as Wilson coefficients of dimension-six operators in the weak effective theory below the electroweak scale [13, 14].

Several building blocks necessary for the prediction of $\Gamma_{\text{sl}}$ and the moments to high orders in $\alpha_s$ and the $1/m_b$ expansion have been presented over the last 30 years (see Sec. 3.2 for an exhaustive list of references). The `kolya` library provides the first comprehensive open-source framework in which all available corrections are implemented and validated. A schematic overview of the perturbative corrections implemented for the total rate and the moments is given in Tab. 1. This document accompanies the first release of `kolya` and details the specifics of the code. Although this paper represents a reference for future analyses of Belle II measurements of inclusive $B \rightarrow X_c l \bar{\nu}_l$ decays and gives a first outlook of `kolya` with basic examples to try in a Jupyter notebook, it is not meant to be a review article on semileptonic decays. To obtain a deeper understanding of the scientific part, the user is referred to e.g. Refs. [15–17]. The software `kolya` complements in scope several other open-source packages in HEP, in particular `flavio` [18], `EOS` [19], `HEPfit` [20], `HAMMER` [21] and `SuperIso` [22].

This article is structured as follows. In Sec. 2, we present the definitions of observables. Their implementation in the code is discussed in Sec. 3 where we discuss various ingredients implemented and quote the original references from which the material was obtained. The definition of the effective Hamiltonian parametrizing NP effects is given in Sec. 4. Section 5 focuses on basic usage of the code, illustrating the installation, the classes implemented in the code, use of the code for calculating $\Gamma_{\text{sl}}$ and the moments together with details about our validation of the code. We close in Sec. 6 with an outlook.

## 2 Definitions

We focus on the the inclusive semileptonic decay of a $B$ meson,

$$\overline{B}(p_B) \rightarrow X_c(p_X) \, l(p_l) \, \bar{\nu}_l(p_\nu) \text{ with } l = e, \mu \,, \tag{1}$$

in its rest frame where $p_B = (M_B, \vec{0})$. We assume that the charged lepton and the neutrino are massless. Let $q = p_l + p_\nu$ be the sum of the neutrino and the charged lepton momenta. The total momentum of the hadronic system $X_c$ is denoted by $p_X = p_B - q$. The energy of $e$ or $\mu$ in the rest-frame of the $B$ meson is $E_l$.

In the HQE framework, predictions can be made for various differential rates with respect to the total leptonic energy $q_0 = E_l + E_\nu$, the charged lepton's energy $E_l$, and the dilepton invariant mass $q^2$. However, these predictions cannot be directly compared to data on a point-by-point basis, as the phase space region allowed at the parton level is smaller than the physical one. Moreover, close to the endpoint, the $1/m_b$-suppressed terms contain singularities. Consequently, the inclusive process

$\bar{B} \to X_c l \bar{\nu}_l$ must be compared to theoretical predictions of integrated quantities, such as the total rate

$$\Gamma_{sl} = \int \frac{d^3\Gamma}{dq^2 \, dq_0 \, dE_l} dq^2 \, dq_0 \, dE_l \,, \tag{2}$$

or partial decay rate with a lower cut on $q^2$ or $E_l$:

$$\Delta\Gamma_{sl}(q_{cut}^2) = \int_{q^2 \geq q_{cut}^2} \frac{d\Gamma}{dq^2} dq^2 \,, \qquad\qquad \Delta\Gamma_{sl}(E_{cut}) = \int_{E_l \geq E_{cut}} \frac{d\Gamma}{dE_l} dE_l \,. \tag{3}$$

The moments of normalized differential distributions can also be calculated in the framework of the HQE. They are defined by

$$\langle (O)^n \rangle_{cut} = \int_{cut} (O)^n \frac{d\Gamma}{dO} dO \Big/ \int_{cut} \frac{d\Gamma}{dO} dO \,, \tag{4}$$

where $O$ strands for $E_l, q^2$ or $M_X^2$, and $d\Gamma/dO$ is the spectrum for the observable $O$. The term "cut" in the integrals refers to a selection threshold on $q^2$ or on $E_l$. Theoretically, the dependence of the moments on such a lower cut offers additional insights into the HQE parameters, improving their extraction in the global fits. Experimentally, the full spectrum is often not measurable due to detector acceptance. For instance, at the $B$-factories electrons are selected with $E_l \geq E_{cut}$ and $E_{cut} \simeq 0.5$ GeV to suppress the background.

Starting from $n = 2$, one normally compares data for centralized moments. For the charged lepton energy spectrum, the centralized moments are

$$\ell_1(E_{cut}) = \langle E_l \rangle_{E_l \geq E_{cut}} \,, \qquad\qquad \ell_n(E_{cut}) = \left\langle (E_l - \langle E_l \rangle)^n \right\rangle_{E_l \geq E_{cut}} \text{ for } n \geq 2 \,. \tag{5}$$

For the hadronic invariant mass spectrum, they are given by

$$h_1(E_{cut}) = \langle M_X^2 \rangle_{E_l \geq E_{cut}} \,, \qquad\qquad h_n(E_{cut}) = \left\langle (M_X^2 - \langle M_X^2 \rangle)^n \right\rangle_{E_l \geq E_{cut}} \text{ for } n \geq 2 \,. \tag{6}$$

The first moment is the mean value of the differential distribution over the considered range and the second moment is its variance. Higher moments, e.g. $n = 3$ and $n = 4$, are also measured. In addition, the measurement of the $q^2$ moments, defined by

$$q_1(q_{cut}^2) = \langle q^2 \rangle_{q^2 \geq q_{cut}^2} \,, \qquad\qquad q_n(q_{cut}^2) = \left\langle (q^2 - \langle q^2 \rangle)^n \right\rangle_{q^2 \geq q_{cut}^2} \text{ for } n \geq 2 \,. \tag{7}$$

was suggested in Ref. [23]. It was shown that $q^2$ moments are invariant under reparametrization and therefore depending on a reduced set of HQE parameters (which we will introduce in Sec. 3), like the total rate (see Refs. [23, 24]). The definition of the $q^2$ moments do not incorporate a cut $E_l > E_{cut}$ since it would break reparametrization invariance (RPI). In order to preserve RPI, the selection $q^2 > q_{cut}^2$ was proposed. This also implies an indirect constraint on the energy of the charged lepton through

$$E_l \geq \frac{M_B^2 + q_{cut}^2 - M_D^2 - \lambda^{1/2}(M_B^2, q_{cut}^2, M_D^2)}{2M_B} \,, \tag{8}$$

where $\lambda(a, b, c) = a^2 + b^2 + c^2 - 2ab - 2ac - 2bc$ is the Källén function. Therefore, a cut on $q^2$ can effectively exclude low-energy electrons in the experimental analysis, just as a cut on $E_l$ would.

# 3 Implementation in the SM

## 3.1 Building blocks

We use the heavy quark expansion (HQE) and express the total semileptonic width $\Gamma_{sl}$ and the kinematic moments as a double expansion in $1/m_b$ and $\alpha_s$. While working with the HQE, it is often advantageous to consider dimensionless quantities normalized w.r.t. the bottom quark mass $m_b$. We will denote them with a "hat(^)": e.g. $\hat{q}^2 = q^2/m_b^2, \hat{E}_l = E_l/m_b$.

As a starting point, we define the moments

$$Q_{ij} = \frac{1}{\Gamma_0} \int_{\text{cut}} dE_l \, dq_0 \, dq^2 \, (q^2)^i (q_0)^j \, \frac{d^3\Gamma}{dq^2 \, dq_0 \, dE_l} \, , \tag{9}$$

where $E_l$ is the lepton energy, $q_0 = (E_l + E_\nu) = v \cdot q$ is the total leptonic energy with $v = p_B/M_B$ and $q^2$ is the leptonic invariant mass. Schematically, we write

$$Q_{ij} = (m_b)^{2i+j} \left[ Q_{ij}^{(0)} + Q_{ij}^{(1)} \frac{\alpha_s(\mu_s)}{\pi} + Q_{ij}^{(2)} \left( \frac{\alpha_s(\mu_s)}{\pi} \right)^2 + \frac{\mu_\pi^2}{m_b^2} \left( Q_{ij,\pi}^{(0)} + Q_{ij,\pi}^{(1)} \frac{\alpha_s(\mu_s)}{\pi} \right) \right.$$

$$+ \frac{\mu_G^2(\mu_b)}{m_b^2} \left( Q_{ij,G}^{(0)} + Q_{ij,G}^{(1)} \frac{\alpha_s(\mu_s)}{\pi} \right) + \frac{\rho_D^3(\mu_b)}{m_b^3} \left( Q_{ij,D}^{(0)} + Q_{ij,D}^{(1)} \frac{\alpha_s(\mu_s)}{\pi} \right)$$

$$\left. + \frac{\rho_{LS}^3(\mu_b)}{m_b^3} \left( Q_{ij,LS}^{(0)} + Q_{ij,LS}^{(1)} \frac{\alpha_s(\mu_s)}{\pi} \right) + O\left( \frac{1}{m_b^4} \right) \right], \tag{10}$$

where $\Gamma_0 = m_b^5 G_F^2 A_{\text{ew}} |V_{cb}|^2/(192\pi^3)$ and $G_F$ is the Fermi constant. The factor

$$A_{\text{ew}} = \left( 1 + \frac{\alpha}{\pi} \log \frac{M_Z}{\mu_b} \right)^2 \tag{11}$$

stems from short-distance radiative corrections at the electroweak scale [25], where $M_Z$ is the mass of the $Z$ boson and $\mu_b$ is a low-energy scale of the order of the bottom mass. The strong coupling constant $\alpha_s \equiv \alpha_s^{(n_f)}(\mu_s)$ is taken with $n_f$ active quarks and at the renormalization scale $\mu_s$. In kolya, we adopt $\alpha_s^{(4)}(\mu_s)$ as the expansion parameter.

To leading order in $1/m_b$, the heavy $B$ meson decay coincides with the decay of a free bottom quark computed in perturbative QCD. Starting from $O(1/m_b^2)$, the predictions depend on a set of HQE parameters: non-perturbative matrix elements of local operators. These are denoted by $\mu_\pi^2, \mu_G^2, \rho_D^3, \rho_{LS}^3$. The tree-level expressions are known also to higher orders in $1/m_b$ (see Refs. [23, 26–29]). They are implemented in kolya up to $1/m_b^5$. However, in (10) they are omitted to keep a compact notation. The explicit definitions of the HQE parameters up to $1/m_b^5$ are reported in Appendix A. The HQE parameters in (10) are quoted in what we refer to as the "historical" basis employed in e.g. Refs. [30, 31]. For RPI quantities, like $q^2$ moments, it is however useful to work in the RPI basis, which has a reduced number of parameters [23, 32]. The differences between these two bases are detailed in Refs. [32, 33].

The functions denoted by $Q_{ij}$ are the fundamental building blocks necessary to assemble the predictions for the centralized moments $q_i$ in (7) and $h_i$ in (6). The functions at the partonic level are denoted by $Q_{ij}^{(n)}$, where $n$ corresponds to the power of $\alpha_s$ which multiplies it. Similarly, we denote with $Q_{ij,G}^{(n)}$ the coefficients of $\mu_G^2$, $Q_{ij,\pi}^{(n)}$ the coefficients of $\mu_\pi^2$, etc.

All functions depend on the mass ratio

$$\rho \equiv \frac{m_c}{m_b} , \tag{12}$$

where $m_c$ and $m_b$ refer to the on-shell masses of the charm and bottom quark. In (9), the subscript "cut" refers to certain restrictions in the phase-space integration. For the prediction of $q_i$ in (7), we apply the cut $q^2 > q_{\text{cut}}^2$ so that various build blocks in (10) depend on $\rho$ and $\hat{q}_{\text{cut}}^2$: $Q_{ij} \rightarrow Q_{ij}(\rho, \hat{q}_{\text{cut}}^2)$. For the hadronic moments $h_n$ in (6), the restriction is on the electron energy $E_l > E_{\text{cut}}$, so that the building blocks are functions of $\rho$ and $\hat{E}_{\text{cut}}$: $Q_{ij} \rightarrow Q_{ij}(\rho, \hat{E}_{\text{cut}})$ (see (18) for the relation between $h_n$ and $Q_{ij}$).

The QCD corrections depend explicitly on the renormalization scale $\mu_s$ of the strong coupling constant starting at $O(\alpha_s^2)$. Starting from $O(\alpha_s)$, the functions $Q_{i,j,G}, Q_{i,j,D}$ and $Q_{i,j,LS}$ depend on the additional scale $\mu_b$ corresponding to the scale at which the Wilson coefficients of the HQET Lagrangian are matched onto QCD. In (10), we refrained to explicitly write the dependence on $q_{\text{cut}}^2$, $E_{\text{cut}}$ and all other scales to keep the notation compact.

To construct the centralized electron energy moments in (5), we consider the moments of the charged-lepton energy $E_l = p_l \cdot v$ within the HQE:

$$\begin{aligned}
L_i &= \frac{1}{\Gamma_0} \int_{E_l \geq E_{\text{cut}}} dE_l \, dq_0 \, dq^2 \, (E_l)^i \frac{d^3\Gamma}{dq^2 \, dq_0 \, dE_l} \\
&= (m_b)^i \Bigg[ L_i^{(0)} + L_i^{(1)} \frac{\alpha_s(\mu_s)}{\pi} + L_i^{(2)} \left( \frac{\alpha_s(\mu_s)}{\pi} \right)^2 + \frac{\mu_\pi^2}{m_b^2} \left( L_{i,\pi}^{(0)} + L_{i,\pi}^{(1)} \frac{\alpha_s(\mu_s)}{\pi} \right) \\
&\quad + \frac{\mu_G^2(\mu_b)}{m_b^2} \left( L_{i,G}^{(0)} + L_{i,G}^{(1)} \frac{\alpha_s(\mu_s)}{\pi} \right) + \frac{\rho_D^3(\mu_b)}{m_b^3} \left( L_{i,D}^{(0)} + L_{i,D}^{(1)} \frac{\alpha_s(\mu_s)}{\pi} \right) \\
&\quad + \frac{\rho_{LS}^3(\mu_b)}{m_b^3} \left( L_{i,LS}^{(0)} + L_{i,LS}^{(1)} \frac{\alpha_s(\mu_s)}{\pi} \right) + O\left( \frac{1}{m_b^4} \right) \Bigg] ,
\end{aligned} \tag{13}$$

where in this case we allow for a cut on $E_l$ only. The labeling of the functions $L_{ij}^{(n)}, L_{ij,G}^{(n)}, L_{ij,\pi}^{(n)}$ etc., follow the same rules as for the $Q_{ij}$ moments. The functions $L_i$ depend on $\rho$ and the cut $\hat{E}_{\text{cut}}$: $L_i \rightarrow L_i(\rho, \hat{E}_{\text{cut}})$.

The total semileptonic rate corresponds to

$$\Gamma_{\text{sl}} = \frac{m_b^5 G_F^2 A_{\text{ew}}}{192\pi^3} |V_{cb}|^2 Q_{0,0}(\rho, 0) = \Gamma_0 L_0(\rho, 0) , \tag{14}$$

with no cut applied, namely $E_{\text{cut}} = q_{\text{cut}}^2 = 0$. For the partial decay width, we similarly have

$$\Delta\Gamma_{\text{sl}}(E_{\text{cut}}) = \Gamma_0 L_0(\rho, \hat{E}_{\text{cut}}) , \qquad\qquad \Delta\Gamma_{\text{sl}}(q_{\text{cut}}^2) = \Gamma_0 Q_{0,0}(\rho, \hat{q}_{\text{cut}}^2) . \tag{15}$$

The ratios defined in (4) correspond to

$$\langle (q^2)^n \rangle = \frac{Q_{n,0}}{Q_{0,0}} , \qquad\qquad \langle E_l^n \rangle = \frac{L_n}{L_0} . \tag{16}$$

The centralized moments are obtained by inserting the double expansions of (10) or (13) into (4-7) and re-expanding in $\alpha_s$ and $1/m_b$ up to the relevant order. To assemble the $M_X^2$ moments, we express the hadronic invariant mass in terms of the parton level quantities in the $B$ rest-frame:

$$M_X^2 = (M_B v - q)^2 = M_B^2 + q^2 - 2M_B q_0 \,, \tag{17}$$

where $M_B$ is the $B$ meson mass. The moments of $M_X^2$ are obtained as linear combinations of the mixed moments $Q_{i,j}$:

$$M_n = \frac{1}{\Gamma_0} \int_{E_l \geq E_{\text{cut}}} dE_l \, dq_0 \, dq^2 (M_B^2 - 2M_B q_0 + q^2)^n \frac{d^3\Gamma}{dE_l \, dq_0 \, dq^2}$$

$$= \sum_{i=0}^{n} \sum_{j=0}^{i} \binom{n}{i}\binom{i}{j} (M_B^2)^{n-i} (-2M_B)^{i-j} Q_{j,i-j} \,, \tag{18}$$

and $\langle (M_X^2)^n \rangle = M_n/M_0$.

In kolya, we first implement all building blocks introduced above, corresponding to the on-shell scheme for both $m_b$ and $m_c$. We collect in the Tab. 1 the list of references from which the various building blocks are retrieved. The implementation of the building blocks is described in Sec. 3.2. The implementation of the NNLO corrections to $E_l$ and $M_X$ moments, based on the results published in Ref. [34], requires a dedicated discussion in Sections 3.3 and 3.4. In Sec. 3.6, we perform a scheme change to the kinetic scheme to obtain the final prediction for the total rate and the centralized moments.

## 3.2 Analytic expressions and grids for QCD corrections

In kolya, the tree-level expressions up to $O(1/m_b^5)$ (see Refs. [23, 26–29]) are implemented in an exact analytic form. For example, the tree-level expressions at leading order in $1/m_b$ for $L_i^{(0)}(\rho, \hat{E}_{\text{cut}})$ in (13) are coded in Python as follows.

```python
from numba import jit
import math

@jit(cache=True, nopython=True)
def L_0(i,elcuthat,r,dEl,dr):
    """ tree-level (partonic) for El moments and their derivatives """
    y  = 2*elcuthat
    logy = math.log((1-y)/r**2)
    # tree-level function
    if (dEl == 0 and dr == 0 and i==0):
        return (1-8*r**2-6*r**4+12*logy*r**4+4*r**6-r**8
            -(2*r**6)/(-1+y)**2-(6*r**4*(1
            +r**2))/(-1+y)
            -2*r**4*(-3+r**2)*y+2*(-1+r**2)*y**3+y**4)
    if (dEl == 0 and dr == 0 and i==1):
        return (3*logy*r**4*(3+r**2)+(7-75*r**2-180*r**4
            +120*r**6-15*r**8+3*r**10)/20)
```

| $\Gamma_{\rm sl}$ | tree | $\alpha_s$ | $\alpha_s^2$ | $\alpha_s^3$ |
|---|---|---|---|---|
| Partonic | | [35] | [36–39] | [40] |
| $\mu_\pi^2, \mu_G^2$ | [1, 2] | [41–44] | | |
| $\rho_D^3, \rho_{LS}^3$ | [45] | [46] | | |
| $1/m_b^4, 1/m_b^5$ | [23, 26–29] | | | |

| $q_n(q_{\rm cut}^2)$ | tree | $\alpha_s$ | $\alpha_s^2$ |
|---|---|---|---|
| Partonic | | [46, 47] | [48, 49] |
| $\mu_G^2, \mu_\pi^2$ | [1, 2] | [42, 43] | |
| $\rho_D^3, \rho_{LS}^3$ | [45] | [46] | |
| $1/m_b^4, 1/m_b^5$ | [23, 28, 29] | | |

| $\ell_n(E_{\rm cut}), h_n(E_{\rm cut})$ | tree | $\alpha_s$ | $\alpha_s^2 \beta_0$ | $\alpha_s^2$ |
|---|---|---|---|---|
| Partonic | | [47, 50, 51] | [47] | [34]* |
| $\mu_G^2, \mu_\pi^2$ | [1, 2] | [43, 52] | | |
| $\rho_D^3, \rho_{LS}^3$ | [45] | | | |
| $1/m_b^4, 1/m_b^5$ | [26–29] | | | |

Table 1: Schematic overview of the perturbative corrections implemented for the rate $\Gamma_{\rm sl}$, the $q^2$ moments, the $E_l$ and $M_X^2$ moments. (*) The $\alpha_s^2$ corrections to $h_n$ and $\ell_n$ are only available for several $\rho$ and $E_{\rm cut}$ values in Ref. [34].

```
        -r**6/(-1+y)**2-(r**4*(3+5*r**2)))/(-1+y)
        +6*r**4*y-(r**4*(-3+r**2)*y**2)/2
        +(3*(-1+r**2)*y**4)/4+(2*y**5)/5)
    ...
```

where the arguments of L_0 refer to the moment $i \in [0, \ldots, 4]$, the normalized electron energy cut $\hat{E}_{\text{cut}} = E_{\text{cut}}/m_b$ (elcuthat) and the mass ratio $\rho = m_c/m_b$ indicated by (r). The additional two arguments (dEl,dr) are positive integers referring to the derivatives of $L_i(\rho, \hat{E}_{\text{cut}})$ w.r.t. to $\hat{E}_{\text{cut}}$ or $\rho$. These derivatives are required when expressing the predictions in the kinetic scheme (see detailed discussion in Sec. 3.6). The tree-level expressions up to $O(1/m_b^3)$ for the moments are implemented in the files Q2moments_SM.py, Elmoments_SM.py and MXmoments_SM.py. The power corrections of order $1/m_b^4$ and $1/m_b^5$ are given in separate files Q2moments_HO.py, Elmoments_HO.py and MXmoments_HO.py. For the $q^2$ moments, the higher power corrections in the RPI basis are found in Q2moments_HO_RPI.py

In order to use kolya for phenomenological applications, for instance to perform a fit, it is important to ensure adequate speed for the numerical evaluations. To this end, our implementation utilizes the library Numba [53], which translates Python functions to optimized machine code at run-time using the standard LLVM compiler library. The functions decorated with @jit, as shown in the example above, are compiled to machine code "just-in-time" for execution at native machine-code speed. Numba-compiled routines in Python approach the speed of C or FORTRAN.

Let us now discuss the implementation of the QCD corrections in kolya. For $\Gamma_{\text{sl}}$, the functions in (10) depend only on the mass ratio $\rho$ and we implement the NLO corrections up to $1/m_b^3$ using the analytic expressions given in Ref. [46]. At NNLO in the free bottom quark approximation, there are asymptotic expansions available either in the limit $\rho \to 0$ [36, 37] or $\delta = 1 - \rho \to 0$ [38]. Recently, analytic expressions for the NNLO corrections written in terms of iterated integrals were presented in Ref. [39]. In kolya, we use the expressions expanded in terms of $\delta = 1 - \rho$ up to $\delta^{46}$ given in Ref. [39]. For the third-order correction to $\Gamma_{\text{sl}}$ we implement the asymptotic expansion up to $\delta^{12}$ computed in Ref. [40]. The total rate is implemented up to $1/m_b^3$ in TotalRate_SM.py, the higher power corrections are given in TotalRate_HO.py and TotalRate_HO_RPI.py.

We use interpolation grids to implements most of the QCD corrections for the moments. Specifically, we use grids for all NLO corrections, the NNLO corrections to the $q^2$ moments and the so-called BLM corrections (of order $\alpha_s^2 \beta_0$) [54] to the $E_l$ and $M_X^2$ moments.

Analytic expressions for the differential rate $d\Gamma/dq^2$ for a free bottom quark are available at NLO from Ref. [55] and at NNLO from Ref. [48] (see also Ref. [49] for a recent independent evaluation of the NNLO corrections). The NLO corrections to leading order in $1/m_b$ were computed also for the triple differential rate in Refs. [47, 50]. The NLO corrections to $\mu_G^2$ and $\rho_D^3$ for the $q^2$ spectrum have been presented in Ref. [46]. By making use of reparametrization invariance [24], one can also show that in the on-shell scheme

$$-\frac{1}{2}Q_{i0}^{(n)}(\rho, q_{\text{cut}}^2) = Q_{i0,\pi}^{(n)}(\rho, q_{\text{cut}}^2)\,, \qquad\qquad -Q_{i0,G}^{(n)}(\rho, q_{\text{cut}}^2) = Q_{i0,LS}^{(n)}(\rho, q_{\text{cut}}^2)\,, \qquad (19)$$

to all orders $n \geq 0$ in the perturbative expansion. Therefore, the functions $Q_{ij}$ entering (10) can be computed for any $q_{\text{cut}}^2$ and $\rho$ via:

$$Q_{i0}(\rho, \hat{q}_{\text{cut}}^2) = \frac{1}{\Gamma_0} \int_{\hat{q}_{\text{cut}}^2}^{(1-\rho)^2} (\hat{q}^2)^i \frac{d\Gamma}{d\hat{q}^2}\, d\hat{q}^2\,. \qquad (20)$$

The differential rate $d\Gamma/d\hat{q}^2$ at higher orders is expressed in terms of functions defined via iterated integrals like harmonic polylogarithms (HPLs) [56] and generalized polylogarithms (GPLs) [57, 58]. It is not convenient to integrate the differential rate numerically "on-the-fly" since there are several HPLs and GPLs whose evaluation (for instance with GiNaC [59]) is time-consuming. For this reason, we opt to implement all higher QCD corrections for the moments not in an exact form, but through Chebyshev two-dimensional grids. The functions implemented via grids are $Q_{ij}^{(1)}, Q_{ij}^{(2)}, Q_{ij,\pi}^{(1)} Q_{ij,G}^{(1)}, Q_{ij,D}^{(1)}, Q_{ij,LS}^{(1)}$ in (10) and $L_{ij}^{(1)}, L_{ij}^{(2)}, L_{ij,\pi}^{(1)} L_{ij,G}^{(1)}$ in (13).

Let us briefly review here how a generic function $f(x)$ can be discretized on a grid consisting of the so-called Chebyshev points (for more details see e.g. Ref. [60]). The idea is to evaluate $f(x)$ in $n$ points $x_n$ corresponding to the zeros of the Chebyshev polynomial $T_n(x)$ of degree $n$:

$$x_k = \cos\left(\frac{\pi(k-1/2)}{n}\right) \text{ for } k = 1, \ldots, n .\tag{21}$$

If $f(x)$ is an arbitrary function defined on the domain $x \in [-1, 1]$, we calculate the coefficients $c_j$, with $j = 0, ..., n-1$ given by

$$c_j = \frac{2}{n}\sum_{k=1}^{n} f(x_k)T_j(x_k) ,\tag{22}$$

which can be employed to construct the polynomial

$$\tilde{f}(x) = -\frac{1}{2}c_0 + \sum_{k=1}^{n-1} c_k T_k(x) \approx f(x) ,\tag{23}$$

approximating $f(x)$ in the interval $[-1, 1]$. In particular $\tilde{f}(x) = f(x)$ for all $n$ zeros of $T_n(x)$. In case the function to interpolate $f(y)$ is defined between two arbitrary limits, e.g. $y \in [a, b]$, we apply the variable transformation

$$x = \frac{y - \frac{1}{2}(b+a)}{\frac{1}{2}(b-a)} ,\tag{24}$$

and then perform the interpolation in $x$ as before.

In our setup, the functions to interpolate depend on $\rho$ and $\hat{q}_{\text{cut}}^2$ (or $\rho$ and $\hat{E}_{\text{cut}}$) and can be obtained via two consecutive one-dimensional Chebyshev interpolations. First, we discretize the interval $\rho \in [1/6, 1/3]$ (relevant for the phenomenology) in $n_\rho$ points distributed according to (21). Then for each $\rho_k \in \{\rho_1, \ldots, \rho_{n_\rho}\}$, we discretize $\hat{q}_{\text{cut}}^2$ or $\hat{E}_{\text{cut}}$ into further $n_{\text{cut}}$ points within the allowed range: $\hat{q}_{\text{cut}}^2 \in [0, (1-\rho)^2]$ or $\hat{E}_{\text{cut}} \in [0, (1-\rho^2)/2]$. An example of how the discretization is performed is shown in Fig. 1, for a grid in $\rho$ and $\hat{q}_{\text{cut}}^2$ with $n_\rho = n_{\text{cut}} = 10$.

To estimate the function at a new point $P = (\rho_P, \hat{q}_P^2)$ (the green diamond in Fig. 1), we proceed as follows. For each $\rho_k, k = 1, \ldots, n_\rho$, we calculate $Q(\rho_k, \hat{q}_P^2(1-\rho_P)^2/(1-\rho_k)^2)$ using one-dimensional interpolations in the variable $\hat{q}_{\text{cut}}^2$. These values calculated at fixed $\rho$ are shown by black crosses in Fig. 1. Afterwards, we use them as nodes for a second interpolation this time along the $\rho$ direction, as displayed by the black dashed line in Fig. 1. The second interpolation yields the final estimate for $f(\rho_P, \hat{q}_P^2)$.

For the implementation of the QCD corrections to $h_i$ and $\ell_i$, which depend on $\rho$ and $E_{\text{cut}}$, we also use Chebyshev interpolation grids. At NLO, it is possible to write the differential rate $d\Gamma/dE_l$ in a closed analytic form at NLO [51] for a free quark. To compute $Q_{ij}(\rho, \hat{E}_{\text{cut}})$ and $L_i(\rho, \hat{E}_{\text{cut}})$ at $O(\alpha_s)$ we

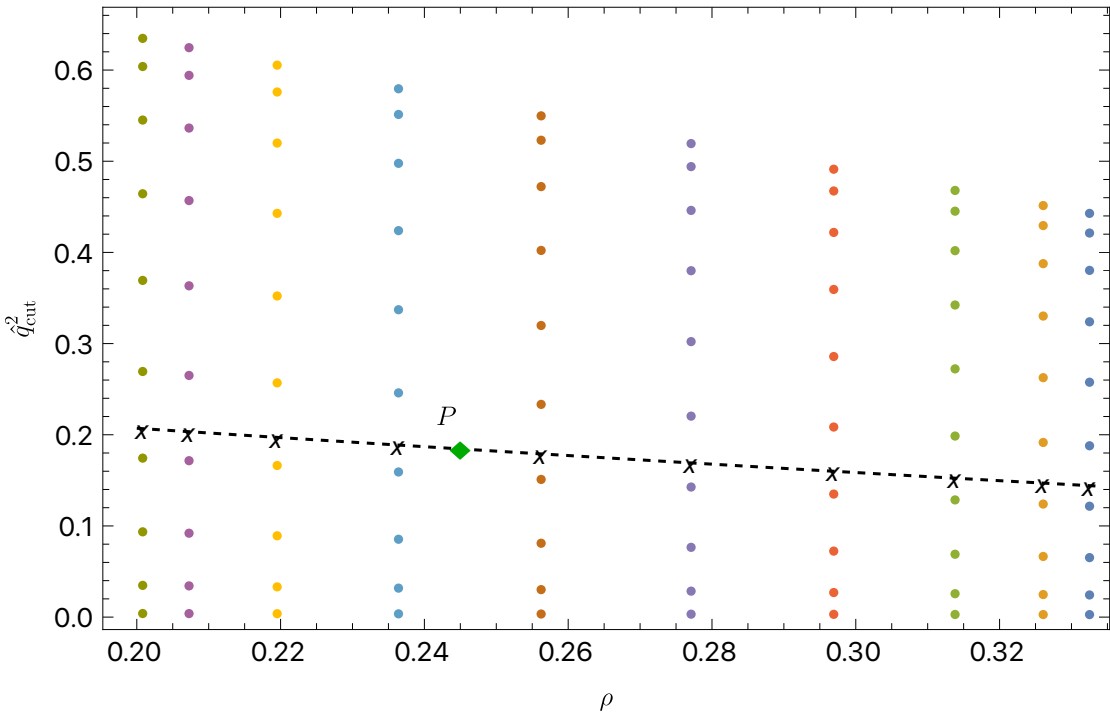

Figure 1: Example of how we discretize functions depending on two variables using Chebyshev nodes. In the plot, we consider the grid for $Q_{ij}$ depending on $\rho$ and $\hat{q}^2_{\text{cut}}$.

perform the one variable integration, as for instance

$$L_i(\rho, \hat{E}_{\text{cut}}) = \frac{1}{\Gamma_0} \int_{\hat{E}_{\text{cut}}}^{(1-\rho^2)/2} (\hat{E}_l)^i \frac{\mathrm{d}\Gamma}{\mathrm{d}\hat{E}_l} \, \mathrm{d}\hat{E}_l \, ,$$

$$Q_{ij}(\rho, \hat{E}_{\text{cut}}) = \frac{1}{\Gamma_0} \int_{\hat{E}_{\text{cut}}}^{(1-\rho^2)/2} \frac{\mathrm{d}Q_{ij}}{\mathrm{d}\hat{E}_l} \, \mathrm{d}\hat{E}_l \, . \tag{25}$$

In the last equation, we define

$$\frac{\mathrm{d}Q_{ij}}{\mathrm{d}\hat{E}_l} = \int (q^2)^i (q_0)^j \frac{\mathrm{d}^3\Gamma}{\mathrm{d}q^2 \, \mathrm{d}q_0 \, \mathrm{d}El} \, \mathrm{d}q^2 \mathrm{d}q_0 \, , \tag{26}$$

which can also be computed analytically up to NLO following Refs. [51, 61].

At order $1/m_b^2$, the NLO corrections $Q_{ij}(\rho, \hat{E}_{\text{cut}})$ and $L_i(\rho, \hat{E}_{\text{cut}})$ can be calculated from the triple differential distributions given in Refs. [42, 43] by performing the phase space integration numerically as described in Ref. [62]. The NLO corrections to $\rho_D^3$ and $\rho_{LS}^3$ are not known at the moment.

The values of the coefficients $c_j$ in (22) for all grids are stored in the directory `grids` as multidimensional arrays. The routines which perform the interpolation of the NLO and NNLO corrections at $O(1/m_b^0)$ are implemented in `NLOpartonic.py` and `NNLOpartonic.py`. The routines for the NLO corrections to the power-suppressed terms are given in `NLOpw.py`.

We stress that all grids are implemented for values of the mass ratio $\rho = [1/6, 1/3]$, which covers the possible values of $m_b$ and $m_c$ of phenomenological relevance. Within this range of $\rho$, the grids

reproduce the original functions with very good accuracy for all $q_{\text{cut}}^2$ and $E_{\text{cut}}$ that are kinematically allowed.

To quantify the accuracy of the interpolation grids we generated 100 random points in the two-dimensional plane $(\rho, \hat{q}_{\text{cut}}^2)$ or $(\rho, \hat{E}_{\text{cut}})$. For each point, we compared the approximation provided by the grids and high-precision evaluations obtained with Mathematica. We verify that the two evaluations differ by less than $10^{-5}$. Outside the range $\rho = [1/6, 1/3]$, an error message is returned to the user that the grids are not provided for that specific (unphysical) value of $\rho$.

### 3.3 NNLO corrections to the lepton energy moments

The NNLO corrections to the $\ell_i$ moments are not known in a closed form. As discussed, the BLM corrections are implemented through interpolation grids. The remaining "non-BLM" corrections are only known for specific values of $\rho$ and $E_{\text{cut}}$ from Ref. [34]. Their functional form can be obtained from a two-dimensional fit to these data points. In order to perform this fit, we write:

$$
\ell_n(\rho, E_{\text{cut}}) = (m_b)^n \Bigg[ Y_n^{(0)} + Y_n^{(1)} \frac{\alpha_s(\mu_s)}{\pi} + \left( \beta_0 Y_n^{(2,\text{BLM})} + Y_n^{(2,\text{nonBLM})} \right) \left( \frac{\alpha_s(\mu_s)}{\pi} \right)^2
$$
$$
+ O\left( \frac{1}{m_b}, \left( \frac{\alpha_s(\mu_s)}{\pi} \right)^3 \right) \Bigg] , \tag{27}
$$

where the $E_{\text{cut}}$- and $\rho$-dependence of $Y_n^{(i)}$ is implied and $Y_n^{(0)}$ is the partonic contribution without any $\alpha_s$ or $1/m_b$ corrections. In terms of the building blocks defined in (13), we can write the non-BLM terms as:

$$
Y_1^{(2,\text{nonBLM})} = \frac{L_1^{(2,\text{nonBLM})}}{L_0^{(0)}} - \frac{L_0^{(2,\text{nonBLM})} L_1^{(0)}}{(L_0^{(0)})^2} ,
$$

$$
Y_2^{(2,\text{nonBLM})} = \frac{L_2^{(2,\text{nonBLM})}}{L_0^{(0)}} - 2\frac{L_1^{(2,\text{nonBLM})} L_1^{(0)}}{(L_0^{(0)})^2} + L_0^{(2,\text{nonBLM})} \left( 2\frac{(L_1^{(0)})^2}{(L_0^{(0)})^3} - \frac{L_2^{(0)}}{(L_0^{(0)})^2} \right) ,
$$

$$
Y_3^{(2,\text{nonBLM})} = \frac{L_3^{(2,\text{nonBLM})}}{L_0^{(0)}} - 3\frac{L_2^{(2,\text{nonBLM})} L_1^{(0)}}{(L_0^{(0)})^2} + 3 L_1^{(2,\text{nonBLM})} \left( 2\frac{(L_1^{(0)})^2}{(L_0^{(0)})^3} - \frac{L_2^{(0)}}{(L_0^{(0)})^2} \right)
$$
$$
+ L_0^{(2,\text{nonBLM})} \left( -6\frac{(L_1^{(0)})^3}{(L_0^{(0)})^4} + 6\frac{L_1^{(0)} L_2^{(0)}}{(L_0^{(0)})^2} - \frac{L_3^{(0)}}{(L_0^{(0)})^2} \right) . \tag{28}
$$

Ref. [34] gives the $L_n^{(2,\text{nonBLM})}$ terms at $\rho = \{0.20, 0.22, 0.24, 0.25, 0.26, 0.28\}$ and $y = \{0, 0.1, \ldots, 0.7\}$, with $y \equiv 2\hat{E}_{\text{cut}}$. From these[1], the non-BLM contributions $Y_n^{(2,\text{nonBLM})}$ to the $\ell_n$ moments are obtained by combining with the tree-level building blocks $L_n^{(0)}$. In Ref. [63], these non-BLM contributions are studied in detail and compared to the effect of their BLM counterparts. We fit the values for $Y_n^{(2,\text{nonBLM})}$ assuming the following polynomial ansatz

$$
Y_n(\rho, y) = \sum_{i=1}^{5} (a_{n,i} + b_{n,i}\rho)(y + \rho^2 - 1)^i , \tag{29}
$$

---

[1]Note that the $L_i^{(n)}$ defined in Ref. [34] are normalized to the total partonic rate without cut while we only normalize to $\Gamma_0$ defined in (14).

for each moment $n$. Following Ref. [63], we only include one power of $\rho$ in our ansatz, but keep up terms up to $y^5$ in our interpolating fit. In addition, the ansatz is chosen to ensure that the non-BLM corrections vanish at the end point $y = 1 - \rho^2$ [34]. We stress that our approach differs from Ref. [63] as we first construct $Y_n^{(2,\text{nonBLM})}$ in each available $(\rho, y)$ point and then perform the analysis. Fitting first the $L_n^{(2,\text{nonBLM})}$ and then combining them resulted in strongly oscillating functions due to accidental cancellations. Fitting directly $Y_n$, we find

$$Y_1^{(2,\text{nonBLM})}(\rho, y) =$$

$$(72.57\rho - 25.62)(y + \rho^2 - 1)^5 + (177.60\rho - 64.65)(y + \rho^2 - 1)^4$$

$$+ (157.19\rho - 59.27)(y + \rho^2 - 1)^3 + (62.69\rho - 24.32)(y + \rho^2 - 1)^2$$

$$+ (11.25\rho - 4.35)(y + \rho^2 - 1) ,$$

$$Y_2^{(2,\text{nonBLM})}(\rho, y) =$$

$$(12.61 - 44.47\rho)(y + \rho^2 - 1)^5 + (32.28 - 112.34\rho)(y + \rho^2 - 1)^4$$

$$+ (29.36 - 100.36\rho)(y + \rho^2 - 1)^3 + (11.07 - 36.89\rho)(y + \rho^2 - 1)^2$$

$$+ (1.42 - 4.55\rho)(y + \rho^2 - 1) ,$$

$$Y_3^{(2,\text{nonBLM})}(\rho, y) =$$

$$(32.23\rho - 7.28)(y + \rho^2 - 1)^5 + (79.34\rho - 17.72)(y + \rho^2 - 1)^4$$

$$+ (67.06\rho - 14.67)(y + \rho^2 - 1)^3 + (21.52\rho - 4.49)(y + \rho^2 - 1)^2$$

$$+ (1.71\rho - 0.289)(y + \rho^2 - 1) . \tag{30}$$

These functions are implemented in `kolya`.

In Fig. 2, we show our fit results for $Y_{1,2,3}^{(2,\text{nonBLM})}$, as a function of $y$ for fixed $\rho = 0.22$ (solid blue line). Due to the fit ansatz, we observe a light oscillatory behavior as a function of $y$. In black, we show the constructed data points at fixed $y$ obtained from Ref. [34]. The fit uncertainty is given by the red dotted line, which represents the 90% C.L. interval of the fit. Since we only have data points up to $y = 0.7$ and impose that the contribution vanishes at the endpoint, we notice large uncertainties towards higher $y$ values. In a typical phenomenological analysis, missing higher order terms (here $\alpha_s^3$) terms would be accounted for by varying the scale of $\alpha_s$. The blue bland corresponds to the effect of such a scale variation between $[\alpha_s(m_b^{\text{kin}}/2), \alpha_s(2m_b^{\text{kin}})]$. For $Y_1$, we observe that the $\alpha_s$ variation covers the data points and fit uncertainty. For the higher moments, we observe that the fit uncertainty is higher than the $\alpha_s$ variation for large $y$. However, given the smallness of the non-BLM contributions to these moments, our default setting is to not include an additional uncertainty for these corrections. We stress, however, that the implementation can easily be adjusted to included either an additional overall uncertainty for the $Y_i^{2,\text{nonBLM}}$ functions or to adjust the fit functions entirely to account for the issues described above.

### 3.4 NNLO corrections to the hadronic invariant mass moments

The BLM corrections to the hadronic moments can be obtained from Ref. [47].

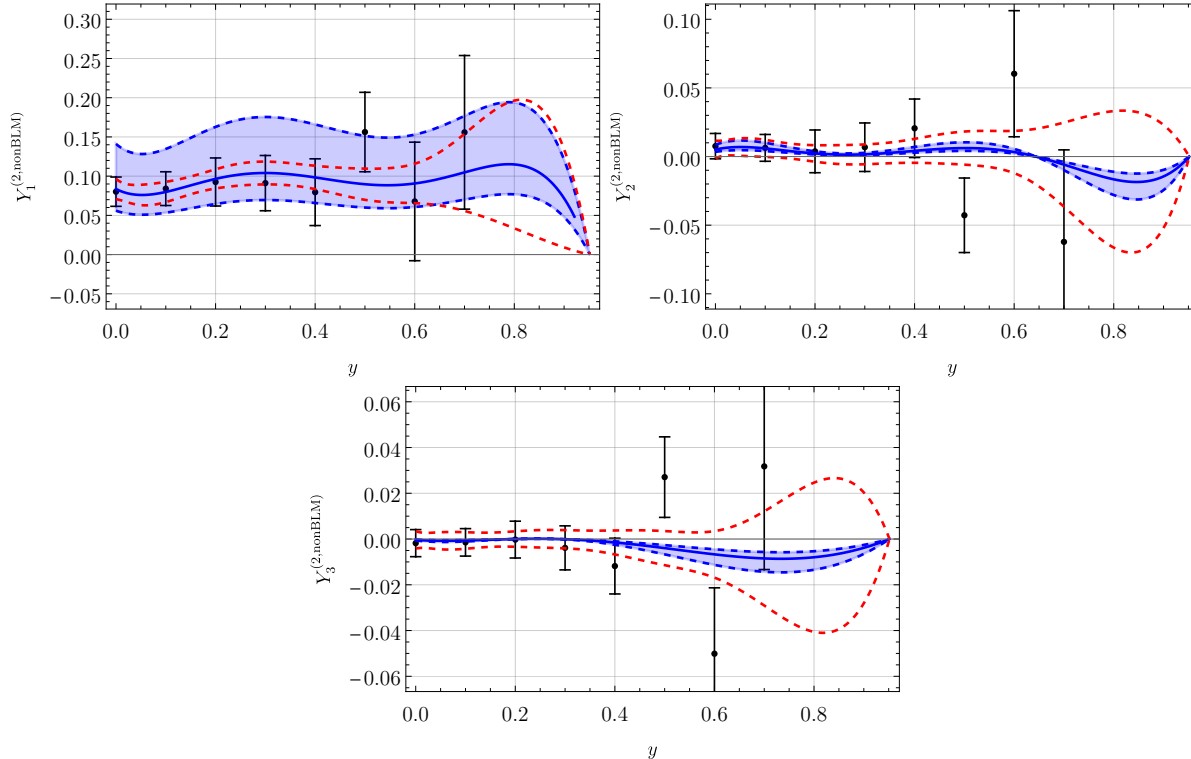

Figure 2: Results of our fitted non-BLM contributions $Y_n^{(2,\mathrm{nonBLM})}$ (solid blue line) as a function of $y = 2E_{\mathrm{cut}}/m_b$ for $\rho = 0.22$ as discussed in the text. Data points are constructed from Ref. [34].

The only available "non-BLM" calculations are for fixed $\rho$ and $E_{\text{cut}}$ and for specific mixed moments defined as [34]:

$$H_{ij}(\rho, \hat{E}_{\text{cut}}) = \frac{1}{\Gamma_0 L_0^{(0)}(\rho, 0)} \int_{E_{\text{cut}}} d\hat{E}_l \, d\hat{q}_0 \, d\hat{q}^2 \left( \frac{m_x^2 - m_c^2}{m_b^2} \right)^i \left( \frac{E_h}{m_b} \right)^j \frac{d^3\Gamma}{d\hat{E}_l \, d\hat{q}_0 \, d\hat{q}^2} \, , \qquad (31)$$

where $L_0^{(0)}(\rho, 0)$ is defined in (13) and where $m_x$ and $E_x$ are the partonic invariant mass and energy. Our building blocks $Q_{ij}$ defined in (18) can obtained by taking the relevant linear combination of the $H_{ij}$ functions. Finally, these can then be used to extract the centralized $M_X^2$ moments $h_1, h_2$ and $h_3$ defined in (6). It is straightforward to show the non-BLM corrections provided in Ref. [34] are not sufficient to obtain the corrections to $h_2$ and $h_3$, as also noted in Ref. [63]. Specifically, in the notation of (31) this would require the non-BLM corrections to $H_{20}, H_{30}, H_{21}$, which are not provided in Ref. [34] and are currently unknown.

To determine the effect of the non-BLM terms on $h_1$, we write

$$h_n(y, \rho) = X_n(y, \rho) + m_b^2 \left( \frac{\alpha_s(\mu_s)}{\pi} \right)^2 \left[ \beta_0 X_n^{(2,\text{BLM})}(y, \rho) + X_n^{(2,\text{nonBLM})}(y, \rho) \right] \, , \qquad (32)$$

where $X_n^{(2,\text{nonBLM})}$ is the non-BLM contribution, $X_n^{(2,\text{BLM})}$ the BLM contribution and $X_n$ contains all other contributions. Since we can only obtain the non-BLM corrections for specific points in $\rho$ and $E_{\text{cut}}$, we proceed by making a fit to obtain a functional form for the non-BLM corrections, as done for the lepton energy moments. Here, we use

$$X_1^{(2,\text{nonBLM})}(\rho, y) = \sum_{i=0}^{5} (a_{1,i} + b_{1,i}\rho) y^i \, , \qquad (33)$$

as our fit ansatz, where we do not require that non-BLM contributions to the first moment vanishes at the end-point. Using the data points from Ref. [34], we then find

$$\begin{aligned} X_1^{(2,\text{nonBLM})}(y, \rho) = \, & y^5(33.13 - 97.49\rho) + y^4(224.82\rho - 71.16) \\ & + y^3(49.01 - 160.90\rho) + y^2(37.72\rho - 10.92) \\ & + y(0.73 - 2.56\rho) + 2.08\rho - 0.92 \, . \end{aligned} \qquad (34)$$

We note that in Ref. [34], no uncertainties are given on the calculated $H_{ij}$ values if they are smaller than 1%. However, to obtain $X_1$, we have to take several linear combinations of the $H_{ij}$ given in Ref. [34], which may lead to large cancellations. Therefore, to be conservative, we do assume a 1% uncertainty on all data points for which no uncertainty was given. Lowering this uncertainty does not change the outcome significantly.

In Fig. 3, we show $X_1^{(2,\text{nonBLM})}$ as a function of $y$. As for the lepton energy moments, the red dotted line shows the fit uncertainty, which now diverges close to the end-point. This is however not an issue, because the experimental data usually does not have lepton energy cuts $y > 0.8$. In addition, we note the small uncertainty on the data points which we set at 1% as discussed above, clearly much smaller than the effect of the $\alpha_s$ variation (shown by the blue band). As such, we do not implement an additional fit uncertainty on (34) and implement this function into kolya. A similar fit was done in Ref. [64], where also $\rho^2$ terms were taken into account. However, the exact functional form of $X_1$ was

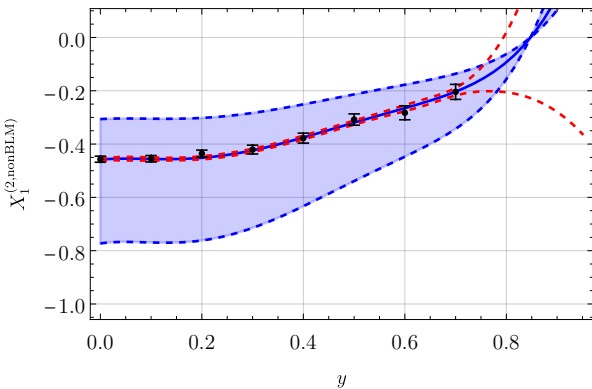

Figure 3: The non-BLM contributions $X_1^{(2,\text{nonBLM})}$ as a function of $y = 2E_{\text{cut}}/m_b$ for $\rho = 0.22$ as described in the text.

not given. We do not notice an improvement in the fit when including such terms and therefore use the minimal fit ansatz presented above. As discussed, the non-BLM corrections to $h_2$ and $h_3$ cannot be obtained from Ref. [34]. Therefore, these effects are currently not implemented in `kolya`. However, it is straightforward to include these effects later on or to add an additional uncertainty if one does a phenomenological analysis.

## 3.5 Additional comment on non-BLM corrections to hadronic moments

Even tough the full non-BLM corrections to $h_2$ and $h_3$ are not available for all energy cuts, it is possible to estimate their effect. At $y = E_{\text{cut}} = 0$, we can use the expression for $Q_{ij}$ from Ref. [61], where analytic expression in terms of $\delta = 1 - \rho$ were given, to obtain the full $\alpha_s^2$ corrections to $h_2$ and $h_3$. From these, the non-BLM corrections can then be extracted[2]. To get an idea of the size of the missing non-BLM corrections, it is interesting to determine the relative size of the non-BLM contributions with respect to the BLM corrections at $E_{\text{cut}} = 0$. Defining the different contributions following (32), and using $\rho = 0.25$ for illustration, we find

$$\frac{X_1^{(2,\text{nonBLM})}}{\beta_0 X_1^{(2,\text{BLM})}} = -0.237\,, \quad \frac{X_2^{(2,\text{nonBLM})}}{\beta_0 X_2^{(2,\text{BLM})}} = -0.227\,, \quad \frac{X_3^{(2,\text{nonBLM})}}{\beta_0 X_3^{(2,\text{BLM})}} = -0.241\,. \tag{35}$$

We observed that the ratio is rather consistent for all three moments. We do not quote an uncertainty on these values, which could be obtained from Ref. [61] by considering the effect of the highest power in the $\delta$-expansion. Since additional information on the moments is missing, we could assume that the ratio of non-BLM over BLM corrections remains constant for all lepton-energy cuts. This assumption was first discussed in Ref. [63], where the non-BLM/BLM ratios of the available $H_{ij}$ moments from Ref. [34] were calculated for different values of the energy cut $y$. The authors found that this ratio is indeed rather independent of the lepton-energy cut-off $y$. Assuming that the ratio non-BLM/BLM corrections does not change for different $y$, the BLM corrections can then be used to estimate the non-BLM corrections at all values of $y$. In Ref. [63], this approach was used to obtain the missing $H_{20,21,30}^{(2)}$ moments. Interestingly, using Ref. [61], we can calculate for the first time directly the ratio

---

[2]We remind the reader that the BLM effects to these moments are known at several lepton-energy cuts.

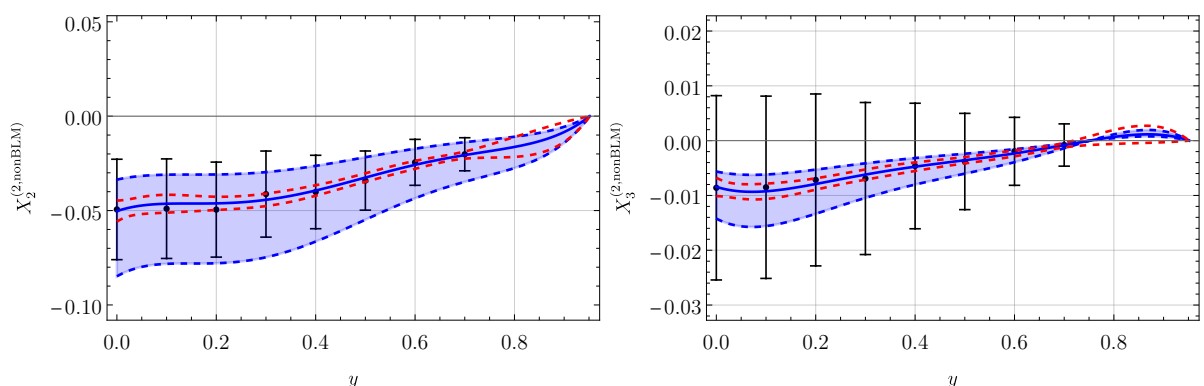

Figure 4: Constructed estimated non-BLM contributions $X_{2,3}^{(2,\text{nonBLM})}$ as a function of $y = 2E_{\text{cut}}/m_b$ for $\rho = 0.22$ obtained by assuming constant non-BLM over BLM corrections as explained in the text.

of $H_{20}^{(2,\text{nonBLM})}/H_{20}^{(2,\text{BLM})}$ and similarly for $\{ij\} = \{21\}, \{30\}$. Doing so, we confirm within uncertainties the ratios quoted in Ref. [63].

For completeness, we also consider the approach of Ref. [63] and assume constant non-BLM over BLM ratios for the missing moments $H_{20,21,30}$. Taking the values for these ratios and their uncertainties from Ref. [63], we find the constructed data points for $X_2^{(2,\text{nonBLM})}$ and $X_3^{(2,\text{nonBLM})}$ given in Fig. 4. In blue, we present the fit to the data using a similar ansatz as in (29) (ensuring also that the contribution vanishes at the end-point). We note that the fit uncertainty (dotted red line) is very small and that also the $\alpha_s$ variation (blue band) is much smaller than the uncertainty on the data points. We observed that $X_3$ is consistent with zero when considering only the calculated data points.

When considering these figures, we should remember that the assumption for the missing moments could also bias these predictions. Specifically, we note that the analytic results in Ref. [48] for the $q^2$ moments with a $q^2$-cut do not show a constant non-BLM over BLM behavior, i.e. the ratio varies with the $q^2$-cut. To conclude, in light of the above discussion, we do not include any non-BLM corrections to the $h_2$ and $h_3$ hadronic mass moments as a default.

Although the above discussion has some caveats, it can still serve as a phenomenological check of the possible effect of not including the $X_2$ and $X_3$ in the hadronic moments. In a phenomenological analysis of $B \to X_c l \bar{\nu}_l$ moments the discussion can serve to justify adding an additional uncertainty on the $h_2$ and $h_3$ moments. As discussed, such an addition can be easily implemented in `kolya`.

## 3.6 Kinetic and $\overline{\text{MS}}$ scheme

The predictions for inclusive decays must be formulated using a short-distance mass scheme for the charm and the bottom quarks, to ensure the cancellation of the leading renormalon divergence [65, 66]. In `kolya`, we adopt the kinetic scheme [5] for the bottom quark mass and the HQE parameters. In the kinetic scheme, we substitute the bottom quark pole mass in favor of the kinetic mass through the relation

$$m_b^{\text{pole}} = m_b^{\text{kin}}(\mu) + [\overline{\Lambda}(\mu)]_{\text{pert}} + \frac{[\mu_\pi^2(\mu)]_{\text{pert}}}{2m_b^{\text{kin}}(\mu)} + O\left(\frac{1}{m_b^2}\right), \tag{36}$$

where the Wilsonian cutoff $\mu$ is a scale chosen such that $\Lambda_{\text{QCD}} \ll \mu \ll m_b$. The last two terms in (36) labeled by "pert" are the HQE parameters calculated in perturbative QCD with the Small Velocity sum rules [67]. The analytic expressions up to $O(\alpha_s^3)$ are given in Refs. [6, 8]. We stress that in our implementation, we follow Ref. [8] and include also the decoupling effects of the charm quark in the kinetic scheme. We show the effect of this decoupling numerically in Sec. 3.7. In the kinetic scheme, also the HQE parameters in (10) and (13) are redefined in the following way:

$$\mu_\pi^2(0) = \mu_\pi^2(\mu) - [\mu_\pi^2(\mu)]_{\text{pert}} , \qquad \rho_D^3(0) = \rho_D^3(\mu) - [\rho_D^3(\mu)]_{\text{pert}} . \qquad (37)$$

These two definitions apply in principle to $\mu_\pi^2$ and $\rho_D^3$ in the $m_b \to \infty$ limit. The analyses from Refs. [9, 30] define the HQE parameters at finite $m_b$ and both setups do not include terms of order $\mu^3$ from the redefinition of $\mu_\pi^2$ and $\mu_G^2$. Ref. [30] and [9] employ different operator bases, which are both implemented in kolya and referred to as the "historical basis" and the "RPI" basis.

For the charm mass, we use the $\overline{\text{MS}}$ scheme. This is related to the pole charm mass via

$$m_c^{\text{pole}} = \overline{m}_c(\mu_c)\left[1 + \frac{\alpha_s(\mu_c)}{\pi}\left(\frac{4}{3} + L\right)\right.$$

$$\left. + \left(\frac{\alpha_s(\mu_c)}{\pi}\right)^2\left(\frac{37}{24}L^2 + \frac{415}{72}L + \frac{779}{96} + \frac{\pi^2}{6} - \frac{\zeta_3}{6} + \frac{1}{9}\pi^2\log(2)\right)\right] + O(\alpha_s^3) , \quad (38)$$

where $L = \log(\mu_c^2/\overline{m}_c^2(\mu_c))$.

In order to implement the scheme change for the quark masses and the HQE parameters, we replace $m_b^{\text{pole}}$ and $m_c^{\text{pole}}$ in the expressions for the centralized moments using (36) and (38) and re-expand the formulas as a series in $\alpha_s$ and $1/m_b$ to the appropriate order. This ensures the cancellation of the leading renormalon order by order. This re-expansion is customarily done in $B \to X_c l \bar{\nu}_l$ analyses, e.g. Refs. [9, 30, 31]. We note that starting at order $\alpha_s^2$, these scheme changes also mix with the NLO corrections. In addition, the scheme changes also induce $\alpha_s \times$ (HQE parameter)-terms. All in all, this makes the scheme change a rather tedious endeavor. The resulting expressions are rather lengthy, therefore we do not hard code all the moments separately, but instead we again make use of building blocks that are common to all moments.

Schematically, we write the mass conversion formulas as follows:

$$m_c^{\text{pole}} = \overline{m}_c(1 + \alpha_s \delta_{m_c,1} + \alpha_s^2 \delta_{m_c,2} + \alpha_s^3 \delta_{m_c,3}) ,$$

$$m_b^{\text{pole}} = m_b^{\text{kin}}(1 + \alpha_s \delta_{m_b,1} + \alpha_s^2 \delta_{m_b,2} + \alpha_s^3 \delta_{m_b,3}) . \qquad (39)$$

The coefficients appearing at order $\alpha_s^n$ from the bottom and charm mass scheme conversion formulas are denoted by $\delta_{m_b,n}$ and $\delta_{m_c,n}$, respectively. Their expressions are given in (36) and (38), respectively. The starting points are the moments in which $m_c$ and $m_b$ enter through $\rho$ as pole scheme masses. The scheme change for a simple function $f(\rho)$ up to $O(\alpha_s)$ then reads schematically

$$f(\rho) = f\left(\frac{m_c^{\text{pole}}}{m_b^{\text{pole}}}\right) = f\left(\overline{m}_c(1 + \alpha_s \delta_{m_c,1})\Big/ m_b^{\text{kin}}(1 + \alpha_s \delta_{m_b,1})\right)$$

$$= f\left(\frac{\overline{m}_c}{m_b^{\text{kin}}}\right) + \alpha_s f'\left(\frac{\overline{m}_c}{m_b^{\text{kin}}}\right)\frac{\overline{m}_c}{m_b^{\text{kin}}}(\delta_{m_c,1} - \delta_{m_b,1}) + O(\alpha_s^2) . \qquad (40)$$

The function $f(\rho)$ and its derivative are the building blocks entering in `kolya`. We note that for the moments, also the $\alpha_s^2$ corrections are taken into account, and for the total rate also $\alpha_s^3$ effects from the scheme change have to be considered. Using building blocks has several advantages with respect to hard coding the expression, especially because of the mixing of these effects with the NLO and NNLO corrections.

In fact for the centralized moments, the role of $f(\rho)$ is played by the building blocks denoted by $Q_{ij}$ and $L_i$ on the r.h.s. of (10) and (13). They appear multiple times after the re-expansion of the ratios in (4) and the scheme change. Therefore, we find it more convenient to calculate first all building blocks in (10) and (13) (and the necessary derivatives) and cache the results at a given value of $\rho$ and kinematical cut. Afterwards, we assemble the centralized moments using expressions similar to (40). This approach yields code with a much smaller size and improved evaluation time.

The coefficients $\delta_{m_b,n}$ and $\delta_{m_c,n}$ which enter in the scheme change are implemented up to $O(\alpha_s^3)$ in the file `schemechange_KINMS.py`. This allows in principle to easily adopt a different mass scheme from our default one, by changing the expression for $\delta_{m_b,n}$ and $\delta_{m_c,n}$ to the required mass scheme in a separate file.

## 3.7 Numerical results for lepton energy and hadronic mass moments

The accuracy of our moment predictions is summarized in Tab. 1. As the NNLO corrections to the $q^2$ moments are known analytically [48], and a detailed discussion was given recently in that reference, we do not discuss these here in detail. However, since our implementation for the lepton energy and hadronic mass moments depends on our fit for the non-BLM contributions, it is interesting to give the relative contributions to the moments. These contributions are given in Tab. 2 for $E_{cut} = 1$ GeV. For the HQE parameters, we use $\mu_\pi^2 = 0.4$ GeV$^2$, $\mu_G^2 = 0.35$ GeV$^2$, $\rho_{LS}^3 = -0.15$ GeV$^3$, $\rho_D^3 = 0.2$ GeV$^3$. For the other input parameters, we employ $m_b^{kin}(1 \text{ GeV}) = 4.563$ GeV, $\overline{m}_c(3 \text{ GeV}) = 0.989$ GeV, $\alpha_s(m_b^{kin}) = 0.2182$. In Tab. 2, we also explicitly give the effect of including the charm decoupling in the kinetic scheme conversion. This contribution is labeled as $\Delta_c$. Our results are in good agreement with Ref. [63] up to the non-BLM corrections. The $\Delta_c$ effects were not included in that reference. Currently, these contributions are automatically included at NNLO in `kolya`, and they cannot be separated from other contributions. We observe that $\Delta_c$ has a 0.1% effect and is similar in size, but opposite in sign as the non-BLM correction. Overall, the $\alpha_s^2$ contributions have a 0.05% effect.

We provide additional numerical examples and comparisons with literature in the GitLab repository, in the Jupyter notebook `example-reproduce_literature.ipynb`.

## 4 Extension to physics beyond the SM

In `kolya`, we also implement NP effects in $b \to c l \bar{\nu}_l$ decays following Ref. [12]. In order to parametrize effects beyond the SM, we use the weak effective theory (WET), an effective field theory valid below the EW scale written as an expansion in powers of the inverse electroweak scale $G_F = 1/(\sqrt{2}v^2)$. The effective Hamiltonian relevant for inclusive semileptonic $B$ decays is

$$\mathcal{H}_{eff} = \frac{4G_F V_{cb}}{\sqrt{2}} \left[ \left(1 + C_{V_L}\right) \mathcal{O}_{V_L} + \sum_{i=V_R,S_L,S_R,T} C_i \mathcal{O}_i \right], \tag{41}$$

| Moment | tree | $\alpha_s$ | $\alpha_s^2 \beta_0$ | $\alpha_s^2$ scheme | $\alpha_s^2$ non-BLM | $\alpha_s^2 \Delta_c$ |
|---|---|---|---|---|---|---|
| $\ell_1$ [GeV] | 1.5650 | 1.5521 | 1.5540 | 1.5459 | 1.5480 | 1.5465 |
| $\ell_2$ [GeV$^2$] | 0.0895 | 0.0870 | 0.0881 | 0.0861 | 0.0867 | 0.0863 |
| $\ell_3$ [GeV$^3$] | -0.0018 | -0.0003 | 0.0004 | 0.0006 | -0.0006 | -0.0006 |
| $h_1$ [GeV$^2$] | 4.166 | 4.331 | 4.304 | 4.417 | 4.381 | 4.403 |
| $h_2$ [GeV$^4$] | 0.609 | 0.818 | 1.001 | 0.987 | - | 0.990 |
| $h_3$ [GeV$^6$] | 5.071 | 4.810 | 4.487 | 4.641 | - | 4.640 |

Table 2: Contributions of the different orders to the moments at $E_{\text{cut}} = 1$ GeV for $\mu_\pi^2 = 0.4$ GeV$^2$ , $\mu_G^2 = 0.35$ GeV$^2$ , $\rho_{LS}^3 = -0.15$ GeV$^3$ , $\rho_D^3 = 0.2$ GeV$^3$. For the other input parameters, we employ $m_b^{\text{kin}}(1 \text{ GeV}) = 4.563$ GeV, $\overline{m}_c(3 \text{ GeV}) = 0.989$ GeV, $\alpha_s(m_b^{\text{kin}}) = 0.2182$.

where we consider only the operators of dimension six contributing to the differential rate at tree-level:

$$\mathcal{O}_{V_{L(R)}} = \left( \bar{c} \gamma_\mu P_{L(R)} b \right) \left( \bar{\ell} \gamma^\mu P_L \nu_\ell \right),$$

$$\mathcal{O}_{S_{L(R)}} = \left( \bar{c} P_{L(R)} b \right) \left( \bar{\ell} P_L \nu_\ell \right),$$

$$\mathcal{O}_T = \left( \bar{c} \sigma_{\mu\nu} P_L b \right) \left( \bar{\ell} \sigma^{\mu\nu} P_L \nu_\ell \right). \tag{42}$$

We define $\sigma^{\mu\nu} = \frac{i}{2}[\gamma^\mu, \gamma^\nu]$ and $P_{L(R)} = 1/2(1 \mp \gamma_5)$. Since only the operator $O_{V_L}$ appears in the SM, we have explicitly factored it out in (41), ensuring that the NP Wilson coefficients vanish in the SM when neglecting electroweak corrections. Taking such effects into account would require putting $C_{V_L} = (\sqrt{A_{\text{ew}}} - 1) + C_{V_L}^{\text{NP}}$, where $A_{\text{ew}}$ is the overall electroweak correction to the total rate defined in (11). We note below that this comment only applies to the branching ratio as $C_{V_L}$ drops out in the normalized moments. The Wilson coefficients $C_i$ are complex numbers and we assume the neutrinos are exclusively left-handed.

Note that below the EW scale, the expansion parameter is $1/\nu$ formally. However, from the point of view of the SMEFT [68], NP effects are associated with an additional expansion parameter $\Lambda$, the scale of new physics above the EW scale. At tree-level, the matching between the SMEFT operators and the operators in the effective Hamiltonian (41) are given in Ref. [69]. From the SMEFT perspective, the Wilson coefficients in (41) are not of order one but additionally suppressed by the small ratio $(\nu/\Lambda)^2$.

The NP contributions to the differential rate of $B \to X_c l \bar{\nu}_l$ have been presented in Ref. [12]. The NP effects for the moments defined in (4) can be written schematically in the following way

$$\langle O \rangle = \xi_{\text{SM}} + |C_{V_R}|^2 \xi_{\text{NP}}^{\langle V_R, V_R \rangle} + |C_{S_L}|^2 \xi_{\text{NP}}^{\langle S_L, S_L \rangle} + |C_{S_R}|^2 \xi_{\text{NP}}^{\langle S_R, S_R \rangle} + |C_T|^2 \xi_{\text{NP}}^{\langle T, T \rangle}$$

$$+ \text{Re}((C_{V_L} - 1)C_{V_R}^*) \xi_{\text{NP}}^{\langle V_L, V_R \rangle} + \text{Re}(C_{S_L} C_{S_R}^*) \xi_{\text{NP}}^{\langle S_L, S_R \rangle} + \text{Re}(C_{S_L} C_T^*) \xi_{\text{NP}}^{\langle S_L, T \rangle}$$

$$+ \text{Re}(C_{S_R} C_T^*) \xi_{\text{NP}}^{\langle S_R, T \rangle}. \tag{43}$$

The terms denoted by $\xi$ are the various interference terms between different operators. They depend on $m_b$, $m_c$, the HQE parameters and $E_{\text{cut}}$ or $q_{\text{cut}}^2$. Ref. [12] provides results for the power corrections at tree-level up to $O(1/m_b^3)$ and NLO perturbative corrections to leading order in the power expansion

$(1/m_b^0)$. In (43), the Wilson coefficients $C_i$ are considered much smaller than one. We therefore expand up to quadratic terms when preparing the predictions for the centralized moments. In the moments, the term $C_{V_L} \xi_{\mathrm{NP}}^{\langle V_L \rangle}$ does not appear because of the normalization. Similarly, the electroweak correction $A_{\mathrm{ew}}$ defined in (11) drops out and we neglect $\alpha \times C_i$ terms. For the branching ratio, the $C_{V_L}$ term appears, resulting in a rescaling of $V_{cb}$. For the branching ratio, the electroweak correction is implemented through $\Gamma_0$ defined above in (11).

The first term $\xi_{\mathrm{SM}}$ in (43) corresponds to the SM prediction, whose implementation has been described in the previous sections. The additional NP contributions $\xi_{\mathrm{NP}}$ generated by the effective operators have been implemented in `kolya` including power corrections up to $1/m_b^3$ and NLO perturbative QCD corrections at partonic level. The implementation closely follows the methods described for the SM case. Namely, we implement the tree-level contributions in an exact form, while for the QCD corrections, we generate interpolation grids for their fast evaluation. The NP contributions to the moments are implemented in `Q2moments_NP.py`, `Elmoments_NP.py` and `MXmoments_NP.py`. The NP extension of the total rate is found in `TotalRate_NP.py`.

## 5 Usage of the library

### 5.1 Installation

The software `kolya` requires Python version 3.6 or above and runs on Linux and Mac. The code is released under the GNU GPL v3 license. The last stable version of the code can be installed from PyPI:

```
$: pip install kolya
```

Alternatively, one can clone the master branch from the GitLab repository via

```
$: git clone https://gitlab.com/vcb-inclusive/kolya.git
```

Afterwards, change the directory to the `kolya` directory and install it with `pip3` in the following way:

```
$: cd kolya
$: pip3 install .
```

The dependencies will be automatically downloaded and installed during the setup. To get started, just import the package into a Python shell or a Jupyter notebook:

```
>>> import kolya
```

Note that the first time `kolya` is loaded, several functions are translated from Python to optimized machine code by Numba and cached. This stage may take from several seconds up to a few minutes.

### 5.2 Parameter classes

The library contains classes to store various real-valued variables. One class is dedicated to the physical parameters like heavy quark masses and the strong coupling constant, one for the HQE parameters, and one for the Wilson coefficients in the NP extension. Dimensionful quantities, like the quark masses, are given in units of GeV. The values of the physical parameters are stored in an object of `parameters.physical_parameters` class

```
>>> par = kolya.parameters.physical_parameters()
```

The new object `par` contains information about $M_B$, $m_b^{\text{kin}}(\mu_{\text{kin}})$, $\overline{m}_c(\mu_c)$ and $\alpha_s^{(4)}(\mu_s)$. The bottom quark mass in the kinetic scheme and its scale are given in the variables `mbkin` and `scale_mbkin`, while the charm mass in the $\overline{\text{MS}}$ at renormalization scale $\mu_c$ corresponds to the variables `mcMS` and `scale_mcMS`. The strong coupling constant $\alpha_s^{(4)}$ and its renormalization scale scale $\mu_s$ correspond to the class variables `alphas` and `scale_alphas`. The class initializes also the renormalization scales of the HQE parameters $\mu_G^2$, $\rho_D^3$ and $\rho_{LS}^3$ through the variables `scale_muG`, `scale_rhoD` and `scale_rhoLS`. At present, these are set equal to $m_b^{\text{kin}}(\mu_{\text{kin}})$. We implement a printable representation of the object in order to inspect the stored values

```
>>> par
bottom mass:       mbkin( 1.0  GeV)      =   4.563  GeV
charm mass:        mcMS( 3.0  GeV)       =   0.989  GeV
coupling constant: alpha_s( 4.563  GeV) =   0.2182
```

One can also use the method `par.show()`, which is introduced to inspect the class with optional arguments. The current default values are based on the latest version of the FLAG 21 review [70] as of February 2024 [71]. Values different from the default ones can be set during initialization. For instance, we can initialize the $\overline{m}_c(2\,\text{GeV}) = 1.094$ GeV as follows:

```
>>> par = kolya.parameters.physical_parameters(mcMS=1.094,scale_mcMS=2.0)
>>> par
bottom mass:       mbkin( 1.0  GeV)      =   4.563  GeV
charm mass:        mcMS( 2.0  GeV)       =   1.094  GeV
coupling constant: alpha_s( 4.563  GeV) =   0.2185
```

The example above shows that during initialization, the values of `mcMS` and `scale_mcMS` must be set consistently. The following command

```
>>> par = kolya.parameters.physical_parameters(mcMS=1.094)
```

would initialize the charm mass to the (unphysical) value $\overline{m}_c(3\,\text{GeV}) = 1.094$ GeV. In order to set the quark masses at scales different from the default ones in a consistent way, we include the method FLAG2024. For instance, we set the quark masses at a scale $\mu_{\text{kin}} = \mu_c = 2$ GeV in the following way:

```
>>> par = kolya.parameters.physical_parameters()
>>> par.FLAG2024(scale_mcMS=2.0, scale_mbkin=2.0)
>>> par
bottom mass:       mbkin( 2.0 GeV)      = 4.295893710310068 GeV
charm mass:        mcMS( 2.0 GeV)       = 1.094393479105018 GeV
coupling constant: alpha_s( 4.563 GeV) = 0.21851076435211256
```

Internally, the bottom and quark masses are recalculated using CRunDec [72–74] using the default values from Ref. [71]. The scale of the strong coupling constant can be modified in a similar way:

```
>>> par = kolya.parameters.physical_parameters()
>>> par.FLAG2024(scale_alphas=3.0)
>>> par
bottom mass:       mbkin(1.0 GeV)      = 4.563512263245122 GeV
charm mass:        mcMS(3.0 GeV)       = 0.989 GeV
coupling constant: alpha_s(3.0 GeV) = 0.25360900216169513
```

Also in this case, we internally use `CRunDec` to evaluate $\alpha_s^{(4)}(3\,\text{GeV})$.

The values of the HQE parameters in the historical basis (sometimes referred to as the "perp" basis in literature) are stored into an object of the class `parameters.HQE_parameters`. By default, their values are set to zero unless explicitly initialized:

```
>>> hqe = kolya.parameters.HQE_parameters(
            muG = 0.306,
            rhoD = 0.185,
            rhoLS = -0.13,
            mupi = 0.477)
>>> hqe
mupi  =   0.477   GeV^2
muG   =   0.306   GeV^2
rhoD  =   0.185   GeV^3
rhoLS =   -0.13   GeV^3
```

where by default only the values up to $1/m_b^3$ are presented to minimize the verbosity: there are many operators at order $1/m_b^4$ and $1/m_b^5$. However, their values can be inspected with the method `show` and the additional options `show(flagmb4=1)` or `show(flagmb5=1)`.

We introduce the class `parameters.HQE_parameters_RPI` for the HQE parameters in the RPI basis:

```
>>> hqe = kolya.parameters.HQE_parameters_RPI(
            muG = 0.306,
            rhoD = 0.185,
            mupi = 0.477)
```

The parameter classes `LSSA_HQE_parameters` and `LSSA_HQE_parameters_RPI` contain numerical values for the HQE parameters in the historical and RPI basis, respectively, obtained using the "lowest-lying state saturation ansatz" (LLSA). The LLSA approximates the higher-order HQE parameters by expressing them through the bottom quark mass $m_b$, the HQE parameters $(\mu_\pi^2)^\perp$ and $(\mu_G^2)^\perp$, and the excitation energies $\epsilon_{1/2}$ and $\epsilon_{3/2}$. For further details on the LLSA, we refer to Ref. [75], and for the expressions of the higher-order HQE parameters and their LLSA values, we refer to Refs. [28, 33].

The Wilson coefficients of the effective dimension-six operators defined in (41) are initialized via the class `parameters.WCoefficients` as complex numbers:

```
>>> wc = kolya.parameters.WCoefficients(SL=0.1+0.1j,SR=-0.05+0.05j)
>>> wc
C_{V_L} =   0
C_{V_R} =   0
C_{S_L} =   (0.1+0.1j)
C_{S_R} =   (-0.05+0.05j)
C_{T} =   0
```

The Wilson coefficients $C_{V_L}, C_{V_R}, C_{S_L}, C_{S_R}$ and $C_T$ are denoted by `VL`, `VR`, `SL`, `SR` and `T` respectively. By default, they are initialized to zero.

### 5.3 Moment predictions

We implemented the first four centralized moments of the $q^2$ spectrum and the first three moments of $E_l$ and $M_X^2$. To evaluate them, we first need to initialize three objects for the physical parameters, the HQE parameters, and the Wilson coefficients:

```
>>> par = kolya.parameters.physical_parameters()
>>> hqe = kolya.parameters.HQE_parameters(
            muG = 0.306,
            rhoD = 0.185,
            rhoLS = -0.13,
            mupi = 0.477)
>>> wc = kolya.parameters.WCoefficients()
```

The prediction for the $q^2$ moments receive as inputs the value of $q_{\text{cut}}^2$ expressed in GeV$^2$, and the three objects par, hqe and wc. The functions Q2moments.moment_n_KIN_MS(q2cut, par, hqe, wc), where n= 1, 2, 3, 4, return the value of each $q^2$ moment. For example, to evaluate $q_1(q_{\text{cut}}^2 = 8.0\,\text{GeV}^2)$, type

```
>>> q2cut = 8.0    #GeV^2
>>> kolya.Q2moments.moment_1_KIN_MS(q2cut, par, hqe, wc)
8.996406491576911 #GeV^2
```

The result is provided in the respective powers of GeV$^{2n}$. The suffix KIN and MS refers to the scheme for bottom (kinetic) and charm ($\overline{\text{MS}}$) masses. By default, the evaluation considers power corrections up to $1/m_b^3$. The corrections of order $1/m_b^4$ and $1/m_b^5$ can be included by setting the optional arguments flagmb4=1 and flagmb5=1. For instance, we can set the higher-order HQE parameters $m_1 = 0.1\,\text{GeV}^4$ and $r_1 = 0.1\,\text{GeV}^5$ and compare the predictions up to order $1/m_b^5$ in the following way:

```
>>> hqe.m1=0.1 #GeV^4
>>> hqe.r1=0.1 #GeV^5
>>> kolya.Q2moments.moment_1_KIN_MS(q2cut, par, hqe, wc)
8.996406491576911 #GeV^2
>>> kolya.Q2moments.moment_1_KIN_MS(q2cut, par, hqe, wc, flagmb4=1)
8.966491804252723 #GeV^2
>>> kolya.Q2moments.moment_1_KIN_MS(q2cut, par, hqe, wc, flagmb4=1, flagmb5=1)
8.75600814926323 #GeV^2
```

Setting flagmb4=0 and flagmb5=0 eliminates all terms of order $1/m_b^4$ and $1/m_b^5$, respectively. We note that at these orders also mixing terms proportional to e.g. $\mu_G^2\mu_\pi^2$ or $\mu_G^2\rho_D^3$ enter which can only be excluded by putting these two flags to zero. Therefore, putting these flags to zero does not have the same effect as simply setting all the $1/m_b^4$ and $1/m_b^5$ HQE elements to zero.

Concerning the perturbative corrections, these are all included by default in the numerical evaluation (see Tab. 1 for the current orders in $\alpha_s$ implemented). For cross-checks with the literature and the study of their impact, the NNLO corrections can be switched off via the optional argument flag_includeNNLO=0 (default flag_includeNNLO=1). NLO corrections to the power-suppressed terms can be excluded with flag_includeNLOpw=0.

Moreover, the option flag_DEBUG=1 will print a report of the various contributions coming from the higher-order QCD corrections:

```
>>> kolya.Q2moments.moment_1_KIN_MS(8.0, par, hqe, wc, flag_DEBUG=1)
Q2moment n. 1 LO =  9.148659808170105
Q2moment n. 1 NLO = api* -1.3195320108322288
Q2moment n. 1 NNLO = api^2* -9.616956957205225
Q2moment n. 1 NLO pw = api* -0.7873907730174552
Q2moment n. 1 NNLO from NLO pw = api^2* 8.390484374123949
```

The contributions denoted by NLO and NNLO are the coefficients in front of $\alpha_s(\mu_s)/\pi$ and $(\alpha_s(\mu_s)/\pi)^2$ to leading order in $1/m_b$. The term NLO pw corresponds to the overall NLO correction in the terms of order $1/m_b^2$ and $1/m_b^3$. In the kinetic scheme, the inclusion of the NLO corrections to the power-suppressed terms induces also an additional $O(\alpha_s^2)$ contribution to leading order in $1/m_b$, which is reported in the last line.

The predictions for the $E_l$ and $M_X^2$ moments follow a similar syntax. The first argument passed to the function corresponds to the value of the cut $E_{\rm cut}$ in units of GeV. For instance, the first moments of $E_l$ and $M_X^2$ for $E_{\rm cut} = 1.0$ GeV are evaluated as follows:

```
>>> par = kolya.parameters.physical_parameters()
>>> hqe = kolya.parameters.HQE_parameters(
            muG = 0.306,
            rhoD = 0.185,
            rhoLS = -0.13,
            mupi = 0.477)
>>> wc = kolya.parameters.WCoefficients()
>>> elcut=1.0 #GeV
>>> kolya.Elmoments.moment_1_KIN_MS(elcut, par, hqe, wc)
1.549812609284058 #GeV
>>> kolya.MXmoments.moment_1_KIN_MS(elcut, par, hqe, wc)
4.348386230761802 #GeV^2
```

Higher moments $\ell_2, \ell_3, h_2, h_3$ are computed in a similar way by replacing `moment_1` with `moment_2` or `moment_3`. The result for $\ell_n(E_{\rm cut})$ is in GeV$^n$, while for $h_n(E_{\rm cut})$ the result is in GeV$^{2n}$.

By default, the moments are calculated using the HQE parameters as defined in the historical basis. The $q^2$ moments and the total rate can also be calculated using the RPI basis adopted in Ref. [9]. The predictions in the RPI basis are obtained by passing the optional argument `flag_basisPERP=0`. In this case, the HQE parameters must be passed to the function through an object of the class `HQE_parameters_RPI`:

```
>>> par = kolya.parameters.physical_parameters()
>>> hqeRPI = kolya.parameters.HQE_parameters_RPI(
            muG = 0.38,
            rhoD = 0.03,
            mupi = 0.43)
>>> wc = kolya.parameters.WCoefficients()
>>> kolya.Q2moments.moment_1_KIN_MS(8.0, par, hqeRPI, wc, flag_basisPERP=0)
9.350141389107412 # GeV^2
```

The RPI basis is supported only for $q_n$ moments and the total rate since reparametrization invariance reduces the number of HQE parameters only for them. Similar to before, the $1/m_b^4$ and $1/m_b^5$ correc-

tions in the RPI basis can be included by using the optional arguments `flagmb4=1` and `flagmb5=1`. For the $\ell_n$ and $h_n$, we stick to the historical basis.

## 5.4 Branching ratio prediction

To obtain the branching ratio $Br(B \to X_c l \bar{\nu}_l)$ or the total semileptonic width $\Gamma_{sl}$, three objects for the physical parameters, the HQE parameters, and the Wilson coefficients must be initialized, as discussed for the moments in the previous subsection. For the total rate $\Gamma_{sl}$ defined in (2), type

```
>>> Vcb = 42.2e-3
>>> kolya.TotalRate.TotalRate_KIN_MS(Vcb, par, hqe, wc)
4.4016320941077224e-14 #GeV
```

where the first argument is the value of $|V_{cb}|$ and the result is expressed in GeV. For the evaluation of the branching ratio, we use

```
>>> Vcb = 42.2e-3
>>> kolya.TotalRate.BranchingRatio_KIN_MS(Vcb, par, hqe, wc)
0.10555834162102022
```

We obtain the branching ratio by dividing $\Gamma_{sl}$ by the average lifetime of the $B^{\pm}$ and $B_0$ mesons.

The partial width $\Delta\Gamma_{sl}(E_{cut})$ with cut on $E_l$ is obtained with

```
>>> Vcb = 42.2e-3
>>> elcut = 1.0 # GeV
>>> kolya.DeltaBR.DeltaRate_KIN_MS(Vcb, elcut, par, hqe, wc)
3.4799361032953045e-14 #GeV
```

where the first argument is the value of $|V_{cb}|$, the second argument the value of $E_{cut}$ in GeV. The result is reported in GeV. The corresponding value for the branching ratio is given by

```
>>> Vcb = 42.2e-3
>>> elcut = 1.0 # GeV
>>> kolya.DeltaBR.DeltaBR_KIN_MS(Vcb, elcut, par, hqe, wc)
0.08345456325227749
```

In the evaluation of the rates and branching ratios described above, we adopt $A_{ew} = 1.014$ for the correction in (11), which corresponds to the scale $\mu_b = m_b^{kin} = 4.563$ GeV. The value of $A_{ew}$ can be modified with the optional argument `Aew=1.014`. For example, it is possible to exclude the contribution of the factor $A_{ew}$ from the calculation of the semileptonic branching ratio in the following way:

```
>>> Vcb = 42.2e-3
>>> kolya.TotalRate.BranchingRatio_KIN_MS(Vcb, par, hqe, wc, Aew=1.0)
0.10410092862033547
```

The functions for the branching ratio and semileptonic (partial) width allow the optional arguments `flagmb4=1` and `flagmb5=1` to include the power corrections of order $1/m_b^4$ and $1/m_b^5$. The predictions in the RPI basis are obtained by passing the optional argument `flag_basisPERP=0`. For cross-checks with the literature and the study of the impact of QCD corrections, the NNLO and N3LO corrections to the total rate can be switched off via the optional arguments `flag_includeNNLO=0` and `flag_includeN3LO=0` (by default, all these corrections are included). Moreover, the effects arising from the NLO corrections to the power-suppressed terms can be excluded with `flag_includeNLOpw=0`.

# 6  Outlook & Conclusion

In this document, we have presented the first version of the open-source library `kolya`, corresponding to the release 1.0. In this release, we have implemented the predictions in the HQE for the total rate and the moments of $q^2$, $E_l$ and $M_X^2$. Currently, this is sufficient for comparison with published experimental results by $B$ factories. We included all higher order corrections in $\alpha_s$ and $1/m_b$ which are available at this specific point in time and are summarized in Tab. 1.

On the GitLab repository, we provide, additionally, interactive tutorials running as a Jupyter notebook and validation notebooks which demonstrate how the library can reproduce the results available in the literature. The library is open source, so code contributions and improvements are very welcome. In particular, new higher-order corrections can be implemented like

- QED effects calculated in Ref. [76],

- exact results for the NNLO corrections to $E_l$ and $M_X^2$ moments with a lower cut $E_{\text{cut}}$,

- renormalization group evolution of the HQE parameters to NLO,

- the NLO corrections in the coefficients of $\rho_D^3$ and $\rho_{LS}^3$ for the $E_l$ and $M_X^2$ moments.

Additional observables can play an important role in better improving the extraction of the HQE parameters or have an important role in testing the SM. These new observables may include

- forward-backward asymmetries $A_{FB}$ and $q^2$, $E_l$ and $M_X^2$ moments for forward and backward events [77, 78],

- the ratio $R_X = \Gamma_{B \to X_c \tau \bar{\nu}_\tau} / \Gamma_{B \to X_c l \bar{\nu}_l}$,

- the lifetime of $B$ mesons within the HQE,

- predictions for the decay into charmless final states $B \to X_u l \bar{\nu}_l$.

Finally, `kolya` could also be extended to include predictions for inclusive $D$ decays discussed in detail in Ref. [79].

## Acknowledgments

We thank F. Bernlochner and M. Prim for ongoing collaboration and suggestions and P. Gambino for providing the results of Refs. [42, 43, 47] in electronic form. We also thank D. Straub for discussion about the Python package `python-rundec` [80], which provides a wrapper around `CRunDec` [72–74]. The work of M.F. is supported by the European Union's Horizon 2020 research and innovation program under the Marie Skłodowska-Curie grant agreement No. 101065445 - PHOBIDE. The work of I.S.M. was supported by the Deutsche Forschungsgemeinschaft (DFG, German Research Foundation) under grant 396021762 – TRR 257 "Particle Physics Phenomenology after the Higgs Discovery". K.K.V. acknowledges support from the project "Beauty decays: the quest for extreme precision" of the Open Competition Domain Science which is financed by the Dutch Research Council (NWO).

# A   Definition of the HQE elements

Here, we define the HQE parameters both in the historical and the RPI basis up to $1/m_b^5$. The conversions between these two bases can be found in Ref. [28].

## A.1   Historical basis

The HQE matrix elements in the historical basis, denoted by "$\perp$", are defined through the spacial covariant derivatives $iD_\mu^\perp = g_{\mu\nu}^\perp iD^\nu$, where

$$g_{\mu\nu}^\perp = g_{\mu\nu} - v_\mu v_\nu \,, \tag{44}$$

as in Refs. [27, 81]. We will employ here the notation $\langle \bar{b}_v \dots b_v \rangle \equiv \langle B(v) | \bar{b}_v \dots b_v | B(v) \rangle$. At $1/m_b^2$, we have

$$2m_B(\mu_\pi^2)^\perp = -\langle \bar{b}_v (iD^\rho)(iD^\sigma) b_v \rangle \, g_{\rho\sigma}^\perp \,,$$

$$2m_B(\mu_G^2)^\perp = \frac{1}{2} \langle \bar{b}_v \left[ (iD^\rho), (iD^\sigma) \right] (-i\sigma^{\alpha\beta}) b_v \rangle \, g_{\rho\alpha}^\perp g_{\sigma\beta}^\perp \,, \tag{45}$$

where $\gamma^\mu \gamma^\nu = g^{\mu\nu} + (-i\sigma^{\mu\nu})$. At $1/m_b^3$, we have

$$2m_B(\rho_D^3)^\perp = \frac{1}{2} \langle \bar{b}_v \left[ (iD^\rho), \left[ (iD^\sigma), (iD^\lambda) \right] \right] b_v \rangle \, g_{\rho\lambda}^\perp v_\sigma \,,$$

$$2m_B(\rho_{LS}^3)^\perp = \frac{1}{2} \langle \bar{b}_v \left\{ (iD^\rho), \left[ (iD^\sigma), (iD^\lambda) \right] \right\} (-i\sigma^{\alpha\beta}) b_v \rangle \, g_{\rho\alpha}^\perp g_{\lambda\beta}^\perp v_\sigma \,. \tag{46}$$

The nine HQE parameters at $1/m_b^4$ were first introduced in Ref. [27]. We list them here:

$$2m_B m_1 = \langle \bar{b}_v (iD^\rho)(iD^\sigma)(iD^\lambda)(iD^\delta) b_v \rangle \frac{1}{3} \left( g_{\rho\sigma}^\perp g_{\lambda\delta}^\perp + g_{\rho\lambda}^\perp g_{\sigma\delta}^\perp + g_{\rho\delta}^\perp g_{\sigma\lambda}^\perp \right) \,,$$

$$2m_B m_2 = \langle \bar{b}_v \left[ (iD^\rho), (iD^\sigma) \right] \left[ (iD^\lambda), (iD^\delta) \right] b_v \rangle \, g_{\rho\delta}^\perp v_\sigma v_\lambda \,,$$

$$2m_B m_3 = \langle \bar{b}_v \left[ (iD^\rho), (iD^\sigma) \right] \left[ (iD^\lambda), (iD^\delta) \right] b_v \rangle \, g_{\rho\lambda}^\perp g_{\sigma\delta}^\perp \,,$$

$$2m_B m_4 = \langle \bar{b}_v \left\{ (iD^\rho), \left[ (iD^\sigma), \left[ (iD^\lambda), (iD^\delta) \right] \right] \right\} b_v \rangle \, g_{\sigma\lambda}^\perp g_{\rho\delta}^\perp \,,$$

$$2m_B m_5 = \langle \bar{b}_v \left[ (iD^\rho), (iD^\sigma) \right] \left[ (iD^\lambda), (iD^\delta) \right] (-i\sigma^{\alpha\beta}) b_v \rangle \, g_{\alpha\rho}^\perp g_{\beta\delta}^\perp v_\sigma v_\lambda \,,$$

$$2m_B m_6 = \langle \bar{b}_v \left[ (iD^\rho), (iD^\sigma) \right] \left[ (iD^\lambda), (iD^\delta) \right] (-i\sigma^{\alpha\beta}) b_v \rangle \, g_{\alpha\sigma}^\perp g_{\beta\lambda}^\perp g_{\rho\delta}^\perp \,,$$

$$2m_B m_7 = \langle \bar{b}_v \left\{ \{ (iD^\rho), (iD^\sigma) \}, \left[ (iD^\lambda), (iD^\delta) \right] \right\} (-i\sigma^{\alpha\beta}) b_v \rangle \, g_{\sigma\lambda}^\perp g_{\alpha\rho}^\perp g_{\beta\delta}^\perp \,,$$

$$2m_B m_8 = \langle \bar{b}_v \left\{ \{ (iD^\rho), (iD^\sigma) \}, \left[ (iD^\lambda), (iD^\delta) \right] \right\} (-i\sigma^{\alpha\beta}) b_v \rangle \, g_{\rho\sigma}^\perp g_{\alpha\lambda}^\perp g_{\beta\delta}^\perp \,,$$

$$2m_B m_9 = \langle \bar{b}_v \left[ (iD^\rho), \left[ (iD^\sigma), \left[ (iD^\lambda), (iD^\delta) \right] \right] \right] (-i\sigma^{\alpha\beta}) b_v \rangle \, g_{\rho\beta}^\perp g_{\lambda\alpha}^\perp g_{\sigma\delta}^\perp \,. \tag{47}$$

Finally, eighteen more parameters are present at $1/m_b^5$, as defined in Ref. [27]:

$$2m_B r_1 = \langle \bar{b}_v (iD_\mu)(ivD)^3 (iD^\mu) b_v \rangle \, ,$$

$$2m_B r_2 = \langle \bar{b}_v (iD_\mu)(ivD)(iD^\mu)(iD)^2 b_v \rangle \, ,$$

$$2m_B r_3 = \langle \bar{b}_v (iD_\mu)(ivD)(iD_\nu)(iD^\mu)(iD^\nu) b_v \rangle \, ,$$

$$2m_B r_4 = \langle \bar{b}_v (iD_\mu)(ivD)(iD)^2 (iD^\mu) b_v \rangle \, ,$$

$$2m_B r_5 = \langle \bar{b}_v (iD)^2 (ivD)(iD)^2 b_v \rangle \, ,$$

$$2m_B r_6 = \langle \bar{b}_v (iD_\mu)(iD_\nu)(ivD)(iD^\nu)(iD^\mu) b_v \rangle \, ,$$

$$2m_B r_7 = \langle \bar{b}_v (iD_\mu)(iD_\nu)(ivD)(iD^\mu)(iD^\nu) b_v \rangle \, ,$$

$$2m_B r_8 = \langle \bar{b}_v (iD_\alpha)(ivD)^3 (iD_\beta)(-i\sigma^{\alpha\beta}) b_v \rangle \, ,$$

$$2m_B r_9 = \langle \bar{b}_v (iD_\alpha)(ivD)(iD_\beta)(iD)^2 (-i\sigma^{\alpha\beta}) b_v \rangle \, ,$$

$$2m_B r_{10} = \langle \bar{b}_v (iD_\mu)(ivD)(iD^\mu)(iD_\alpha)(iD_\beta)(-i\sigma^{\alpha\beta}) b_v \rangle \, ,$$

$$2m_B r_{11} = \langle \bar{b}_v (iD_\mu)(ivD)(iD_\alpha)(iD^\mu)(iD_\beta)(-i\sigma^{\alpha\beta}) b_v \rangle \, ,$$

$$2m_B r_{12} = \langle \bar{b}_v (iD_\alpha)(ivD)(iD_\mu)(iD_\beta)(iD^\mu)(-i\sigma^{\alpha\beta}) b_v \rangle \, ,$$

$$2m_B r_{13} = \langle \bar{b}_v (iD_\mu)(ivD)(iD_\alpha)(iD_\beta)(iD^\mu)(-i\sigma^{\alpha\beta}) b_v \rangle \, ,$$

$$2m_B r_{14} = \langle \bar{b}_v (iD_\alpha)(ivD)(iD)^2 (iD_\beta)(-i\sigma^{\alpha\beta}) b_v \rangle \, ,$$

$$2m_B r_{15} = \langle \bar{b}_v (iD_\alpha)(iD_\beta)(ivD)(iD)^2 (-i\sigma^{\alpha\beta}) b_v \rangle \, ,$$

$$2m_B r_{16} = \langle \bar{b}_v (iD_\mu)(iD_\alpha)(ivD)(iD_\beta)(iD^\mu)(-i\sigma^{\alpha\beta}) b_v \rangle \, ,$$

$$2m_B r_{17} = \langle \bar{b}_v (iD_\alpha)(iD_\mu)(ivD)(iD^\mu)(iD_\beta)(-i\sigma^{\alpha\beta}) b_v \rangle \, ,$$

$$2m_B r_{18} = \langle \bar{b}_v (iD_\mu)(iD_\alpha)(ivD)(iD^\mu)(iD_\beta)(-i\sigma^{\alpha\beta}) b_v \rangle \, . \tag{48}$$

## A.2 RPI basis

The RPI HQE matrix elements up to $1/m_b^4$ have been determined in Ref. [32]. We list them here:

$$2m_B\mu_\pi^2 = -\langle \bar{b}_v \, (iD)^2 \, b_v \rangle \, ,$$

$$2m_B\mu_G^2 = \langle \bar{b}_v \, (iD_\alpha)(iD_\beta)(-i\sigma^{\alpha\beta}) \, b_v \rangle \, ,$$

$$2m_B\tilde{\rho}_D^3 = \frac{1}{2}\langle \bar{b}_v \left[ (iD_\mu), \left[ \left( (ivD) + \frac{1}{2m_b}(iD)^2 \right), (iD^\mu) \right] \right] b_v \rangle \, ,$$

$$2m_B r_G^4 = \langle \bar{b}_v \left[ (iD_\mu), (iD_\nu) \right] \left[ (iD^\mu), (iD^\nu) \right] b_v \rangle \, ,$$

$$2m_B\tilde{r}_E^4 = \langle \bar{b}_v \left[ \left( (ivD) + \frac{1}{2m_b}(iD)^2 \right), (iD_\mu) \right] \left[ \left( (ivD) + \frac{1}{2m_b}(iD)^2 \right), (iD^\mu) \right] b_v \rangle \, ,$$

$$2m_B s_B^4 = \langle \bar{b}_v \left[ (iD_\mu), (iD_\alpha) \right] \left[ (iD^\mu), (iD_\beta) \right] (-i\sigma^{\alpha\beta}) \, b_v \rangle \, ,$$

$$2m_B\tilde{s}_E^4 = \langle \bar{b}_v \left[ \left( (ivD) + \frac{1}{2m_b}(iD)^2 \right), (iD_\alpha) \right] \left[ \left( (ivD) + \frac{1}{2m_b}(iD)^2 \right), (iD_\beta) \right] (-i\sigma^{\alpha\beta}) \, b_v \rangle,$$

$$2m_B s_{qB}^4 = \langle \bar{b}_v \left[ (iD_\mu), \left[ (iD^\mu), \left[ (iD_\alpha), (iD_\beta) \right] \right] \right] (-i\sigma^{\alpha\beta}) \, b_v \rangle \, , \tag{49}$$

We note that we choose to use $\mu_\pi^2$ instead of $\mu_3$ as in Ref. [32], in order to avoid factors of $1/\mu_3$ in the centralized moments. Furthermore, we note that $\tilde{\rho}_D^3$, $\tilde{r}_E^4$, and $\tilde{s}_E^4$ contain their so-called RPI-completion terms as described in Refs. [28, 32]. In Ref. [28], the RPI matrix elements at $1/m_b^5$ have been determined to be:

$$2m_B X_1^5 = \langle \bar{b}_v \left[ (ivD), \left[ (ivD), (iD_\mu) \right] \right] \left[ (ivD), (iD^\mu) \right] b_v \rangle \, ,$$

$$2m_B X_2^5 = \langle \bar{b}_v \left[ (ivD), \left[ (iD_\mu), (iD_\nu) \right] \right] \left[ (iD^\mu), (iD^\nu) \right] b_v \rangle \, ,$$

$$2m_B X_3^5 = \langle \bar{b}_v \left[ (iD_\mu), \left[ (ivD), (iD_\nu) \right] \right] \left[ (iD^\mu), (iD^\nu) \right] b_v \rangle$$

$$2m_B X_4^5 = \langle \bar{b}_v \left[ (iD_\mu), \left[ (iD_\nu), \left[ (iD^\mu), \left[ (ivD), (iD^\nu) \right] \right] \right] \right] b_v \rangle \, ,$$

$$2m_B X_5^5 = \langle \bar{b}_v \left[ (ivD), \left[ (ivD), (iD_\alpha) \right] \right] \left[ (ivD), (iD_\beta) \right] (-i\sigma^{\alpha\beta}) \, b_v \rangle \, ,$$

$$2m_B X_6^5 = \langle \bar{b}_v \left[ (ivD), \left[ (iD_\mu), (iD_\alpha) \right] \right] \left[ (iD^\mu), (iD_\beta) \right] (-i\sigma^{\alpha\beta}) \, b_v \rangle \, ,$$

$$2m_B X_7^5 = \langle \bar{b}_v \left[ (iD_\mu), \left[ (ivD), (iD_\alpha) \right] \right] \left[ (iD^\mu), (iD_\beta) \right] (-i\sigma^{\alpha\beta}) \, b_v \rangle \, ,$$

$$2m_B X_8^5 = \langle \bar{b}_v \left[ (iD_\mu), \left[ (ivD), (iD_\alpha) \right] \left[ (iD^\mu), (iD_\beta) \right] \right] (-i\sigma^{\alpha\beta}) \, b_v \rangle \, ,$$

$$2m_B X_9^5 = \langle \bar{b}_v \left[ (iD_\mu), \left[ (ivD), (iD^\mu) \right] \right] \left[ (iD_\alpha), (iD_\beta) \right] (-i\sigma^{\alpha\beta}) \, b_v \rangle \, ,$$

$$2m_B X_{10}^5 = \langle \bar{b}_v \left[ (iD_\mu), \left[ (ivD), (iD^\mu) \right] \left[ (iD_\alpha), (iD_\beta) \right] \right] (-i\sigma^{\alpha\beta}) \, b_v \rangle \, . \tag{50}$$

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
