# Peer review of "Kolya: an open-source package for inclusive semileptonic B decays"

_SciPost Physics Codebases, doi:SciPost Phys. Codebases 55-r1.0 (2025) , SciPost Phys. Codebases 55 (2025)_

## Round 1 · Referee Report · Danny van Dyk (Referee 1) · 2024-11-4

Strengths

  1. Excellent documentation of the underlying formulas used to evaluate the various observables.
  2. Useful and timely provision of a standard code to predict observables in inclusive B decays.

Weaknesses

  1. Minor weaknesses in the documentation, addressed in the detailed report.

Report

The writeup discussed a novel and highly useful piece of open source software for the calculation of observables in inclusive B meson decay. The physics basis for the code is very well documented. Simple examples are provided as part of the manuscript, with more complex examples available from the Gitlab repository.

I think this paper should be accepted for publication, pending some minor corrections and addendums listed under "requested changes".

Moreover, I have the following comments on the interface / code / package. Addressing the following should not be considered a prerequisite for accepting the paper.

  • kolya is not (yet?) installable from PyPI. I would highly recommend uploading a Python wheel to pypi, to facilitate the installation process.

  • Importing kolya ("import kolya") for the first time takes a very long time (longer than 14min on my laptop (Intel i7-1360P)). Based on the comments in the installation section of the paper, I suspect this is due to the just-in-time compilation using numba. I recommend reconsidering the use of numba and consider the use of cython instead. This would enable building a natively-compiled backend code only once.

  • Inspecting an object, such as return value of "par = kolya.parameters.physical_parameters()" can be achieved using the "show()" method. While useful, it is not very python-like. I highly recommend adding a Jupyter / IPython / Python representation to the classes that the users are going to be exposed most. In Jupyter, this would allow to simply "display(par)", which is done automatically if only the object is invoked at the end of a Jupyter cell.

  • The HQE parameters ("kolya.parameters.HQE_Parameters") do not have sensible default values. I recommend adding a function (similar to FLAG2024 for the general parameter class that provided sensible default values.)

  • I am a bit puzzled by the handling of the Wilson coefficients. Why is Vcb, which is also a parameter in the EFT Lagrangian, not part of the object parametrizing the Lagrangian? (I.e., wc) When I change Vcb, I implicitly change the definition of all Wilson coefficients.

Requested changes

  1. The authors mention in the abstract that kolya ``can be used in an interactive Jupyter notebook session''. This is of course true for most Python libraries. My question is: has it been designed to use in a Jupyter notebook? I think the answer is "no" at the present. I would remove this line from the abstract.

  2. The 2nd sentence below eq. (11) starts with a mathematical symbol.

  3. Below eq. (41), the dependence of the observables on the ratio m_c/m_b is discussed. The trick used to make the expressions more managable is quite neat. I think the text would benefit from a comment on the numerical equivalence of this approach and evaluation f at the value of m_c^pole/m_b^pole as determined from the kinematic / MSbar masses. If there is no numerical equivalencen, it would be useful to understand what precision is achieved by the replacement.

  4. It should be documented that the Wilson coefficient parameters are only parametrizing the BSM contribution to the effective Lagrangian, i.e., it is C_V_L^BSM, and the code uses C_V_L = 1.0 * A_ew + C_V_L^BSM. Crucial question: Is the electroweak correction included for BSM contributions? This should also be documented.

Recommendation

Ask for minor revision

  • validity: top
  • significance: high
  • originality: high
  • clarity: high
  • formatting: excellent
  • grammar: excellent

Author:  Matteo Fael  on 2025-02-07  [id 5194]

(in reply to Report 1 by Danny van Dyk on 2024-11-04)

We would like to thank the referee for carefully reading the manuscript and for recommending publication in SciPost. We thank him also for the suggestions on the code. We also note that we have released a new version of Kolya correcting a mistake in our previous implementation. This error was pointed out by [Ref. 29 [2501.09090]] and subsequently updated in [Ref. 28], on which our implementation is based. In the following we answer the questions of the referee:

Q: kolya is not (yet?) installable from PyPI. I would highly recommend uploading a Python wheel to pypi, to facilitate the installation process.

A: The package has been uploaded to pypi.org and it is now installable from pypi.

Q: Importing kolya ("import kolya") for the first time takes a very long time (longer than 14min on my laptop (Intel i7-1360P)). Based on the comments in the installation section of the paper, I suspect this is due to the just-in-time compilation using numba. I recommend reconsidering the use of numba and consider the use of cython instead. This would enable building a natively-compiled backend code only once.

A: Indeed the first time kolya is imported in python, it requires several minutes to perform the compilation in numba. After that, the library is imported in few seconds because the compiled files are cached. We think this is acceptable since it must be done only once during installation. The possibility to migrate to cython will be considered in a subsequent release.

Q: Inspecting an object, such as return value of `par = kolya.parameters.physical_parameters()` can be achieved using the `show()` method. While useful, it is not very python-like. I highly recommend adding a Jupyter / IPython / Python representation to the classes that the users are going to be exposed most. In Jupyter, this would allow to simply `display(par)`, which is done automatically if only the object is invoked at the end of a Jupyter cell.

A: For the classes in the module `parameters` we have implemented the method "__repr__" to automatically show the content of the object.

Q: The HQE parameters ("kolya.parameters.HQE_Parameters" ) do not have sensible default values. I recommend adding a function (similar to FLAG2024 for the general parameter class that provided sensible default values.)

A: We explicitely did not implement default values for the HQE parameters, because it is important to realize that these parameters are are result of a fit to the $B \to X_c l \nu$ data. In general, we expect Kolya to be used to a) perform fits of that data, in which case no inputs are needed or b) check for the effect of a parameter/show general behavior. In case b) it is important to realize that the result depends on the inputs, therefore we do not provide standard inputs. We stress that in addition, $\mu_G, \mu_\pi$ and $\rho_D$ change also depending on the basis considered and also change everytime a new fit is done (e.g. they differ between the $q^2$ only or full fit). So in summary, we prefer to leave this to the user to fix.

Q: I am a bit puzzled by the handling of the Wilson coefficients. Why is Vcb, which is also a parameter in the EFT Lagrangian, not part of the object parametrizing the Lagrangian? (I.e., wc) When I change Vcb, I implicitly change the definition of all Wilson coefficients.

A: We use the EFT Lagrangian employed also by Straub and Jung in hep-ph/1801.01112. The possibility to normalize the Wilson coefficients w.r.t. $V_{cb}$ is a legittimate choice. Concerning the implementaion in Kolya, we prefers to keep separated the NP parameters in the class of Wilson coefficients (which are all zero in SM case) from the value of $V_{cb}$ which is a SM parameter.

Q: The authors mention in the abstract that kolya can be used in an interactive Jupyter notebook session''. This is of course true for most Python libraries. My question is: has it been designed to use in a Jupyter notebook? I think the answer is "no" at the present. I would remove this line from the abstract.

A: We have removed the last sentence of the abstract.

Q. The 2nd sentence below eq. (11) starts with a mathematical symbol.

A: We have corrected it.

Q. Below eq. (41), the dependence of the observables on the ratio $m_c/m_b$ is discussed. The trick used to make the expressions more managable is quite neat. I think the text would benefit from a comment on the numerical equivalence of this approach and evaluation f at the value of $m_c^{pole}/m_b^{pole}$ as determined from the kinematic / MSbar masses. If there is no numerical equivalencen, it would be useful to understand what precision is achieved by the replacement.

A: We have clarified the text around (40) and (41) to better express the point of (41). Indeed the point of (41) is just to make the expressions more managable. The result is exactly the same as hard-coding the expanded version in kolya. The re-expansion is done such that only terms up to the order we consider are kept, that is, if NNLO corrections are known, we only keep $\alpha_s^2$ corrections coming from the scheme change. Here the function "f" is just a simple example which may contain several powers of $\rho$. In addition, at order $\alpha_s^2$ the re-expansion also induces terms that mix with the NLO corrections.

It is custumary to use short distance masses in the moments for these inclusive decays, calculating in the pole scheme would lead to renormalon ambiguities. In principle, the moments can be calculated in the pole scheme (even in Kolya) precisely because of the building blocks. If one would want to calculate in the pole scheme this would be possible by setting all $\delta_{m_c}$ and $\delta_{m_b}$ to zero (and in addition also the perturbative corrections to the HQE parameters). Finally, we note that the precision achieved by the replacement is up to the order we take into account. For the moments, we drop induced terms of order $\alpha_s^3$ and $\alpha_s^2$ HQE parameter because we do not include N^3LO and higher terms. Taking the effects of the scheme change at this order into account would be inconsistent.

Q: It should be documented that the Wilson coefficient parameters are only parametrizing the BSM contribution to the effective Lagrangian, i.e., it is $C_{V_L}^{BSM}$, and the code uses $C_{V_L} = 1.0 * A_{ew} + C_{V_L}^{BSM}$. Crucial question: Is the electroweak correction included for BSM contributions? This should also be documented.

A: We thank the referee for pointing this out. We have clarified this in the text. The situation is a bit more complicated due to the normalization of the observables. Therefore, $C_{V_L}$ drops out in eq.(43) as explained in the text. This means we also neglect $\alpha C_i$ terms, which is justified as the electroweak corrections for these NP parameters are unknown. For the branching ratio the situation is different and then indeed this term acts as a rescaling of $V_{cb}$ as mentioned. In this case, the electroweak corrections are taken into account through $\Gamma_0$ in a default way. Clearly, this can easily be adjusted by putting in the analysis $C_{V_L} = (\sqrt{A_{ew}}-1) + C_{V_L}^{NP}$ as the referee suggest. We have clarified this in the text.

---

## Round 1 · Referee Report · Anonymous (Referee 2) · 2024-11-29

Strengths

1- The manuscript presents a useful code for flavor physics. 2- It is timely, and a useful addition to existing codes.

Weaknesses

1- The writing could be improved for better clarity and to avoid misunderstandings (see report for details). 2- There are several typographical errors, such as: We will denoted them terms up y^5 incorporate a the cut by consider the effect inclusive decayse is favour of ...

Report

The manuscript presents an open-source program dedicated to the B -> X_c l nu inclusive decay. It calculates the total rate and the kinematic moments within the Standard Model and also includes new physics parametrizations. This work is a useful and timely contribution to the existing tools and can be accepted for publication once the points outlined below are addressed.

Requested changes

-The code presented in this manuscript is very specific, focusing solely on the B -> X_c l nu inclusive decay, and does not cover all semileptonic B decays. For instance, neutral current decays are not included. Therefore, the title: "Kolya: an open-source package for inclusive semileptonic B decays" is misleading and should be revised to accurately reflect the scope of the code. This revision should also be applied to the abstract and throughout the main body of the manuscript.

  • For all the equations, every quantity that appears must be defined, and the corresponding values should be provided where applicable. For example, in Eqs. (9) and (10), it should be specified that the Q_ij​ are the moments, the value of n_f​ (the number of active quarks) should be given, the scale mu_s should be provided, and so on.

  • In Eq. (11), the authors provide a hard-coded value for the short-distance radiative corrections, A_ew = 1.01435, and cite a paper from 1978. First, a correct citation should be provided to clarify the source of this value. Second, it is generally not a good practice to hard-code numerically specific corrections, as this may introduce inconsistencies, particularly with respect to the input values used.

  • On page 6, it is stated that the scale at which the Wilson coefficients of the HQET Lagrangian are matched onto QCD is mu_b​. Is this a typo?

  • The higher-order QCD corrections for the moments are not implemented in their exact form, but instead are approximated using Chebyshev two-dimensional grids, which is useful for fast numerical evaluations. However, could there be regions where this approximation might not work well?

  • In Eqs. (30) and (31), no uncertainty is assigned to the fit functions. What is the reason?

  • The explanation in the last paragraph of page 13 (following Eq. (32)) is unclear.

  • After Eq. (34), it is stated: "to be conservative, we assume a 1% uncertainty on all data points for which no uncertainty was given". This choice seems somewhat arbitrary and not very satisfying in my opinion.

  • At the end, the non-BLM corrections to the h2 and h3 are not included. I found the reasoning for this to be quite confusing.

  • In section 4 for the BSM extensions, the C_i are assumed to be real. Why?

  • Still in Section 4 (BSM extensions), it is unclear what exactly is implemented. From reading this brief section, I assume it refers to a parametrization at the level of the effective Lagrangian. However, for example, what is meant by: "we implement the tree-level contributions in an exact form". What specific expressions are being referred to here? Are any particular new physics models considered?

Recommendation

Ask for major revision

  • validity: high
  • significance: good
  • originality: good
  • clarity: ok
  • formatting: good
  • grammar: reasonable

Author:  Matteo Fael  on 2025-02-07  [id 5193]

(in reply to Report 2 on 2024-11-29)
Category:
answer to question

We would like to thank the referee for carefully reading the manuscript and for recommending publication in SciPost. We thank her/him also for the suggestions on the code. We also note that we have released a new version of Kolya correcting a mistake in our previous implementation. This error was pointed out by [Ref. 29 [2501.09090]] and subsequently updated in [Ref. 28], on which our implementation is based. In the following we answer the questions of the referee:

Q:There are several typographical errors
A: We have corrected the typos pointed out by the referee.

Q: The code presented in this manuscript is very specific, focusing solely on the $B \to X_c l \nu$ inclusive decay, and does not cover all semileptonic B decays. For instance, neutral current decays are not included. Therefore, the title: "Kolya: an open-source package for inclusive semileptonic B decays" is misleading and should be revised to accurately reflect the scope of the code. This revision should also be applied to the abstract and throughout the main body of the manuscript.

A: We revised the abstract accordingly, clarifying that the first version of the library focuses on $B \to X_c l \nu$ decays.
We think that in the main text there is no ambiguity since we clearly state in Eq. 1 that we consider only $B \to X_c l \nu$ decays. We prefer not to modify the title because in the near future we plan to include other kinds of decays calculated within the HQE, like D-meson decays or the lifetimes of B mesons.

Q: For *all* the equations, every quantity that appears must be defined, and the corresponding values should be provided where applicable. For example, in Eqs. (9) and (10), it should be specified that the $Q_{ij}$ are the moments, the value of $n_f$. (the number of active quarks) should be given, the scale $\mu_s$ should be provided, and so on.

A: In the revised version, we specified that $Q_ij$ are the moments. We also clarify that our implementation utilizes $\alpha_s$ with $n_f=4$ as an expansion parameter. The scale of $\alpha_s$, $\mu_s$, on the other hand is not fixed and is a free parameter that the user can choose. We also clarified the notation in Eqs. (10) and (13).

Q: In Eq. (11), the authors provide a hard-coded value for the short-distance radiative corrections, $A_{ew} = 1.01435$, and cite a paper from 1978. First, a correct citation should be provided to clarify the source of this value. Second, it is generally not a good practice to hard-code numerically specific corrections, as this may introduce inconsistencies, particularly with respect to the input values used.

A: We now report in Sec. 3.1 the explicit formula for $A_{ew}$, Eq. (11), without quoting any predetermined value at this stage. We only specify in Sec. 5.4 (Branching ratio prediction) that the default value $A_{ew} = 1.014$ is obtained using $\mu_b = m_b^{kin}$(1 GeV) = 4.563 GeV. In an updated version of the package, we give the user the possibility to modify the value of $A_{ew}$ by passing an optional argument to the functions for the branching ratios. This new option in described in Sec. 5.4.

Q: On page 6, it is stated that the scale at which the Wilson coefficients of the HQET Lagrangian are matched onto QCD is $\mu_b$. Is this a typo?

No, this is correct. In principle the matching scale of the HQET Lagrangian
can be different from the renormalization scale of $\alpha_s$.

Q: The higher-order QCD corrections for the moments are not implemented in their exact form, but instead are approximated using Chebyshev two-dimensional grids, which is useful for fast numerical evaluations. However, could there be regions where this approximation might not work well?

A: We clarified at the end of section 3.2 that the grids are implemented only for $1/6 < m_c/m_b < 1/3$, which is the region compatible with the physical values of $m_c$ and $m_b$. Inside this range, all values of $q^2_{cut}$ and $E_{cut}$ are allowed and we quantified that the grids reproduce the original analytic function with an absolute difference better than $10^{-5}$. Outside the range $1/6 < m_c/m_b < 1/3$ the code gives an error message.

Q: In Eqs. (30) and (31), no uncertainty is assigned to the fit functions. What is the reason?

A: Note in the new version (31) and (30) are now merged into one equation, Eq. (30). Indeed, we opt as a default in Kolya not to include an uncertainty on Eqs. (30). We give a detailed explanation for this in the section below (30), and have added a comment that for phenomological analyses users can add additional uncertainties as they deem necessary based on the discussion in this section.

Q: The explanation in the last paragraph of page 13 (following Eq. (32)) is unclear.

A: We have sharpened the discussion.

Q: After Eq. (34), it is stated: "to be conservative, we assume a 1% uncertainty on all data points for which no uncertainty was given". This choice seems somewhat arbitrary and not very satisfying in my opinion.

A: For the fit we use the results reported in Table 3 of hep-ph/0911.4142. In this table, the BLM functions are reported for certain values of $\rho$ and the energy cut. No uncertainty is quoted in this table (contrary to Table 1 and 2 where Vegas integration errors are given). It is however stated that in those cases the uncertainty from the integration is smaller than 1%.

As such, we opt to be conservative and assign a relative uncertainty of 1% to all these values. We then have to convert the $H_{ij}$ functions of [0911.4142] to the relevant combinations. As stated, this requires taking fine-tuned linear combinations. As such, unaccounted correlations and small uncertainties on $H_{ij}$ functions can cause artificial cancellations. To be conservative, we therefore allow for a somewhat large uncertainty on these points. However, we stress that reducing this uncertainty does not change the outcome of the fit significantly. In addition, we note that the uncertainty on the data points (barely visible in Fig. 3) is much smaller than the variation of $\alpha_s$ (blue band). This is also mentioned in the text. Our approach follows that of [62], but in that reference no functional form for $X_1$ was given, making a direct comparision missing. In fact, our work is the first to give the functional form of $X_1$ (and $Y_i$), which hopefully opens the discussion on these terms to further improve the perturbative side of the predictions. For the time being, we believe our approach in this section is the best in light of the available calculations. The non-BLM contributions to the hadronic moments clearly need to be recalculated. We now take this approach.

Q: At the end, the non-BLM corrections to the h2 and h3 are not included. I found the reasoning for this to be quite confusing.

A: The default in Kolya is indeed to not include these terms because they are not known. They can however be estimated under specific assumptions because we know the corrections at y=0. We have now separated this discussion in a subsection (Sec. (3.5)), because we think it is valuable to the community. As mentioned in the reply above, no functional form for $X_1$ was given previously in litaratue. By the time $h_2$ and $h_3$ are known exactely up to NNLO, the
corrections can simply be added to Kolya. In addition, we encourage others to estimate these effects. And Kolya is flexible, such that others may implement their own estimate for these effects. The new subsection is simply to get an idea how big the neglected terms could be, such that this can be taken in to account in a pheno analysis. As stated above, after many discussions, we believe this is the best treatment at the moment. The non-BLM contributions to the hadronic moments clearly need to be recalculated, we now take this approach. Yet at the same time, we leave it free for other users of Kolya to implement their own function of the non-BLM functions.

Q: In section 4 for the BSM extensions, the $C_i$ are assumed to be real. Why?

In the revised version of the code, we promote the Wilson coefficients to complex numbers. Note that this additional assumption do not induce additional interference terms in Eq. (43) for the considered moments. We modified the class associated to the Wilson coefficients, as discussed in Sec. 5.2 of the revised version.

Q: Still in Section 4 (BSM extensions), it is unclear what exactly is implemented. From reading this brief section, I assume it refers to a parametrization at the level of the effective Lagrangian. However, for example, what is meant by: "we implement the tree-level contributions in an exact form". What specific expressions are being referred to here? Are any particular new physics models considered?

A: The section is brief as this just describes the implementation in Kolya. We refer to Ref. [12] for futher details. We have adjusted the section to make this clearler. The expression (43) is implemented in Kolya. This does not assume any specific NP model because the Wilson coefficients are free paramters. Using Kolya all observables can be calculated for specific values of $C_i$. As we quote below (43), the analytic expressions are taken from Ref [12]. The "exact form" referred to here simply means that also for the NP contribution the tree level expressions are coded in an exact form using the formulas derived in Ref. [12]. The implementation is analogous to the implementation of the SM prediction described at the begining of Sec. 3.2. Similaly, the QCD corrections for the NP effects are again implemented as grids to speed up the evaluation.

---

## Editorial Decision

published